# Exploring the Approximation Capabilities of Multiplicative Neural Networks for Smooth Functions

**Ido Ben-Shaul**                                    *ido.benshaul@gmail.com*
*Department of Applied Mathematics*
*Tel-Aviv University, Israel*
*eBay Research*

**Tomer Galanti**                                    *galanti@mit.edu*
*Department of Brain and Cognitive Sciences*
*Massachusetts Institute of Technology (MIT)*
*Cambridge, MA, USA*

**Shai Dekel**                                    *shaidekel6@gmail.com*
*Department of Applied Mathematics*
*Tel-Aviv University, Israel*

**Reviewed on OpenReview:** *https://openreview.net/forum?id=sWQJfb2GSk*

## Abstract

Multiplication layers are a key component in various influential neural network modules, including self-attention and hypernetwork layers. In this paper, we investigate the approximation capabilities of deep neural networks with intermediate neurons connected by simple multiplication operations. We consider two classes of target functions: generalized bandlimited functions, which are frequently used to model real-world signals with finite bandwidth, and Sobolev-Type balls, which are embedded in the Sobolev Space $\mathcal{W}^{r,2}$. Our results demonstrate that multiplicative neural networks can approximate these functions with significantly fewer layers and neurons compared to standard ReLU neural networks, with respect to both input dimension and approximation error. These findings suggest that multiplicative gates can outperform standard fully-connected layers and potentially improve neural network design.

## 1 Introduction

Deep learning has seen tremendous success in solving a wide range of tasks in recent years, including image classification (He et al., 2016; Dosovitskiy et al., 2021; Zhai et al., 2022), language processing (Vaswani et al., 2017; Devlin et al., 2019; Brown et al., 2020), interacting with open-ended environments (Silver et al., 2016; Arulkumaran et al., 2019), and code synthesis (Chen et al., 2021).

Recent empirical studies have shown that neural networks that incorporate multiplication operations between intermediate neurons (Durbin & Rumelhart, 1989; Urban & van der Smagt, 2016; Trask et al., 2018), such as self-attention layers (Vaswani et al., 2017), dynamic convolutions (Wu et al., 2019) and hypernetworks (Ha et al., 2017; Krueger et al., 2017; Littwin & Wolf, 2019; Littwin et al., 2020), are particularly effective. For example, self-attention layers have been widely successful in computer vision (Dosovitskiy et al., 2021; Zhai et al., 2022) and language processing (Cheng et al., 2016; Parikh et al., 2016; Paulus et al., 2018; Vaswani et al., 2017). It has also been shown that one can achieve reasonable performance with Transformers even without applying non-linear activation functions (Levine et al., 2020). Additionally, hypernetworks, which use multiplication to generate network weights seem to improve the performance of neural networks on various meta-learning tasks (von Oswald et al., 2020; Littwin & Wolf, 2019; Bensadoun et al., 2021). In a different

line of work, (Haber & Ruthotto, 2018; Eliasof et al., 2021) showed that incorporating non-linear interactions can stabilize the training of deep networks, including convolutional and graph neural networks. Despite their practical success, the theoretical reasons behind the effectiveness of multiplicative layers in deep learning remain unclear.

In this work, we study the expressive power of neural networks with multiplication layers. Specifically, we want to evaluate the number of neurons and layers needed to approximate a given function within a given error tolerance using a specific architecture. A classic result in the theory of deep learning, known as the universal approximation property, shows that neural networks can approximate any smooth target function with as few as one hidden layer (Cybenko, 1989; Hornik et al., 1989; Funahashi, 1989; Leshno et al., 1993). However, these papers do not provide specific information about the type of architecture and number of parameters required to achieve a given level of accuracy. This is a crucial question, as a high requirement for these resources could limit the universality of neural networks and explain their limited success in some practical applications.

In a recent paper, (Jayakumar et al., 2020) investigated the role of multiplicative interaction as a framework including various classical and contemporary neural network components, including gating, attention layers, hypernetworks, and dynamic convolutions. The authors conjectured that multiplicative layers are well-suited for modeling conditional computations and proved that networks with multiplicative connections are more expressive than those with ReLU fully-connected layers. However, they did not investigate to what extent the multiplicative networks serve as better function approximators than standard ReLU networks.

Previous work has demonstrated that functions in Sobolev spaces can be approximated by a one-hidden layer neural network with analytic activation functions (Mhaskar, 1996). However, the number of neurons required to approximate these functions with an error of at most $\epsilon$ in the $L_\infty$ norm scales as $\mathcal{O}(\epsilon^{-d/r})$, where $d$ is the input dimension, $r$ is the smoothness degree of the target function, and $\epsilon > 0$ is the error rate. This raises the question of whether the curse of dimensionality, the phenomenon whereby the complexity of a model grows exponentially with the input dimension, is inherent to neural networks.

On the other hand, DeVore et al. (1989) proved that any continuous function approximator that approximates all Sobolev functions of order $r$ and dimension $d$ within error $\epsilon$ requires at least $\Omega(\epsilon^{-d/r})$ parameters in the $L_\infty$ norm. This result meets the bound of Mhaskar (1996) and confirms that neural networks cannot avoid the curse of dimensionality for the Sobolev space when approximating in the $L_\infty$ norm. A key question is whether neural networks can overcome this curse of dimensionality for certain sets of target functions, and what kind of architectures provide the best guarantees for approximating these functions.

To overcome the curse of dimensionality, various studies (Mhaskar et al., 2017; Poggio et al., 2020; Kohler & Krzyżak, 2017; Montanelli & Du, 2019; Blanchard & Bennouna, 2022; Galanti & Wolf, 2020) have investigated the approximation capabilities of neural networks in representing other classes of functions with some additional structural properties or by assuming weaker notions of distance, such as the $L_2$ distance. For example, Mhaskar et al. (2017); Poggio et al. (2020) showed that smooth, compositionally sparse functions with a degree of smoothness $r$ can be approximated with the $L_\infty$ distance up to error $\epsilon$ using deep neural networks with $\mathcal{O}(d\epsilon^{-2/r})$ neurons. Other structural constraints have been applied to functions with structured input spaces (Mhaskar, 2010; Nakada & Imaizumi, 2022; Schmidt-Hieber, 2019), compositions of functions (Kohler & Krzyżak, 2017), piecewise smooth functions (Petersen & Voigtländer, 2017; Imaizumi & Fukumizu, 2018). A different line of research has focused on understanding the types of functions that certain neural network architectures can implement with regularity constraints. For example, E et al. (2021) showed that the space of 2-layer neural networks is equivalent to the Barron space when the size of their weights is restricted. They further showed that Barron functions can be approximated within $\epsilon$ using 2-layer networks with $\mathcal{O}(\epsilon^{-2})$ neurons. Another line of research has considered spectral conditions on the function space, allowing functions to be expressed as infinite-width limits of shallow networks (Barron, 1991; Klusowski & Barron, 2018). In (Blanchard & Bennouna, 2022) they considered the space of Korobov functions, which are functions that are practically useful for solving partial differential equations (PDEs). They showed any Korobov function can be approximated up to error $\epsilon$ in $L_2$ distance with a 2-layer neural network with ReLU activation function with $\mathcal{O}(\epsilon^{-1}\log(1/\epsilon)^{1.5(d-1)+1})$ and with a $\mathcal{O}(\log(d))$-depth network with $\mathcal{O}(\epsilon^{-0.5}\log(1/\epsilon)^{1.5(d-1)+1})$ neurons.

It is generally known that real-world datasets (such as CIFAR10 or ImageNet) often (approximately) lie on low-dimensional manifolds of the input space (Tenenbaum et al., 2000; Belkin & Niyogi, 2003). Several publications (Nakada & Imaizumi, 2022; Schmidt-Hieber, 2019; Chen et al., 2019; Kohler et al., 2019; Suzuki, 2018) studied the adaptivity of deep ReLU networks to the intrinsic dimensionality of data and in particular showed that the curse of dimensionality is avoidable when the data lies on a low-dimensional manifold. The work of (Suzuki & Nitanda, 2019) further showed that deep ReLU networks can overcome the curse of dimensionality even when the input data does not lie on a low-dimensional manifold, as long as the target function is included in some anisotropic Besov space Nikol'skii (1975). In (Shen et al., 2022) the authors showed that general target function classes (such as the Hölder or Lipschitz spaces) may be modeled by a single network, of a certain architecture, such that only a small number of 'intrinsic' parameters are used to approximate each individual function.

In a recent paper, Montanelli & Du (2021) provided approximation estimates for generalized bandlimited functions. These functions are commonly used to model signals that have a finite range of frequencies (e.g., waves, video, and audio signals), which is known as a finite bandwidth. In (Montanelli & Du, 2021), it was shown that any bandlimited function can be approximated in $L_2$ within error $\epsilon$ using a ReLU neural network of depth $\mathcal{O}(\log^2(1/\epsilon))$ with $\mathcal{O}(\epsilon^{-2}\log^2(1/\epsilon))$ neurons.

In this paper, we compare the approximation capabilities of multiplicative neural network architectures with those of standard ReLU networks with respect to the $L_2$ distance. In particular, we prove that a multiplicative neural network of depth $\mathcal{O}(\log(\frac{1}{\epsilon}))$ with $\mathcal{O}(\epsilon^{-2}\log(\frac{1}{\epsilon}))$ neurons can approximate any generalized bandlimited function up to an error of $\epsilon$ (with constants depending on the dimension and on the band). This result represents an improvement compared to the findings in (Montanelli & Du, 2021) for traditional ReLU networks. Additionally, we also study the approximation guarantees of neural networks for approximating functions in Sobolev-Type balls of order $r$. We show that for the same error tolerance $\epsilon$, multiplicative neural networks can approximate these functions with depth $\mathcal{O}(d\epsilon^{-1/r})$ and $\mathcal{O}(d\epsilon^{-(2+1/r)})$ neurons, while standard ReLU neural networks require depth $\mathcal{O}(d^2\epsilon^{-2/r})$ and $\mathcal{O}(d^2\epsilon^{-(2+2/r)})$ neurons. These results demonstrate the superior performance of multiplicative gates compared to standard fully-connected layers. In Table 1 we contrast our new bounds with preexisting bounds on the approximation power of neural networks for the Sobolev space, bandlimited functions, and the Sobolev-Type ball.

| Space | Model | # neurons | Depth | Reference |
|---|---|---|---|---|
| $\mathcal{W}^{r,p}$ | $C^\infty$, non-poly | $\mathcal{O}(\epsilon^{-d/r})$ | $\mathcal{O}(1)$ | (Mhaskar, 1996) |
| $\mathcal{W}^{r,\infty}$ | ReLU | $\mathcal{O}(\epsilon^{-d/r}\log\frac{1}{\epsilon})$ | $\mathcal{O}(\log\frac{1}{\epsilon})$ | (Yarotsky, 2017) |
| Bandlimited functions | ReLU | $\mathcal{O}(\epsilon^{-2}\log^2\frac{1}{\epsilon})$ | $\mathcal{O}(\log^2\frac{1}{\epsilon})$ | (Montanelli & Du, 2021) |
| Bandlimited functions | Multiplicative | $\mathcal{O}(\epsilon^{-2}\log\frac{1}{\epsilon})$ | $\mathcal{O}(\log\frac{1}{\epsilon})$ | This paper |
| $\mathcal{B}_{2r,2} \subsetneq \mathcal{W}^{r,2}$ | ReLU | $\mathcal{O}(d^2\epsilon^{-(2+2/r)})$ | $\mathcal{O}(d^2\epsilon^{-2/r})$ | This paper |
| $\mathcal{B}_{2r,2} \subsetneq \mathcal{W}^{r,2}$ | Multiplicative | $\mathcal{O}(d\epsilon^{-(2+1/r)})$ | $\mathcal{O}(d\epsilon^{-1/r})$ | This paper |
| $\mathcal{B}_{1,1}$ | Sigmodial | $\mathcal{O}(d\epsilon^{-2})$ | $\mathcal{O}(1)$ | (Barron, 1993) |

Table 1: Approximation results for Sobolev $\mathcal{W}^{r,p}$, Bandlimited and $\mathcal{B}_{2r,2}$ functions by ReLU and multiplicative neural networks. The number of neurons and the depth are given in $\mathcal{O}$ notation.

Our findings offer a deeper understanding of the application of deep learning techniques in solving partial differential equations (PDEs) and signal processing. For instance, bandlimited functions are often prevalent in signal processing applications, since the acquisition of data from sensors is frequently in the form of bandlimited functions. In such applications, neural networks are used to approximate a given signal function. Our results show that multiplicative neural networks are a preferable choice for approximating such functions.

In addition, recent papers suggest various approaches for solving PDEs with physics-informed neural networks (Raissi et al., 2019; Karniadakis et al., 2021; Kahana et al., 2022; Ben-Shaul et al., 2023; Bar & Sochen, 2021). In these methods, a deep network is trained to approximate a solution for a PDE by minimizing a loss function associated with the differential equation. For various PDEs, such as the heat equation, it was shown that if the initial condition at time $t = 0$ exists in a specific smoothness space, such as the Sobolev space $\mathcal{W}^{r,p}$ (refer to definition in Section 2.1), then the solution remains in that space for all

$t > 0$. In other scenarios, if the initial condition lies in a Sobolev space $\mathcal{W}^{r,p}$, the solution can be ensured to be within a Sobolev space $\mathcal{W}^{s,p}$, where $s \leq r$. Additionally, for elliptic boundary problems with smooth coefficients over $C^2$ smooth domains, if a weak solution exists, it automatically belongs to $\mathcal{W}^{r,2}$ (Evans, 2010). Based on our findings, we recommend the use of physics-aware multiplicative networks as the preferred architecture for such applications.

## 2 Problem Setup

We are interested in determining how complex (i.e., number of trainable parameters, number of neurons and layers) a model ought to be in order to theoretically guarantee approximation of an unknown target function $f$ up to a given approximation error $\epsilon > 0$.

Formally, we consider a Banach space of functions $\mathcal{V}$ (for example, $L_p([0,1]^d)$), equipped with a norm $\|\cdot\|_{\mathcal{V}}$ (for example, $\|\cdot\|_{L_p([0,1]^d)}$), and a set of target functions $\mathcal{U} \subseteq \mathcal{V}$. We also consider a set of approximators $\mathcal{H}$ and seek to quantify the ability of these approximators to approximate $\mathcal{U}$ using the following quantity

$$d_{\mathcal{V}}(\mathcal{H}, \mathcal{U}) = \sup_{f \in \mathcal{U}} \inf_{\hat{f} \in \mathcal{H}} \|\hat{f} - f\|_{\mathcal{V}},$$

which measures the maximal approximation error for approximating a target function $f \in \mathcal{U}$ using candidates $\hat{f}$ from $\mathcal{H}$. Typically, $\mathcal{H}$ is a parametric set of functions (e.g., neural networks of a certain architecture) and we denote by $\hat{f}_\theta \in \mathcal{H}$ a function that is parameterized by a vector of parameters $\theta \in \mathbb{R}^N$. For simplicity, we avoid writing $\theta$ in the subscript when it is obvious from context.

### 2.1 Target Function Spaces

It is generally impossible to approximate arbitrary target functions using standard neural networks, as demonstrated in Theorem 7.2 in (Devroye et al., 1996). As a result, we often consider specific spaces of target functions that satisfy certain smoothness assumptions in order to obtain non-trivial results. In this work, we focus specifically on target functions $\mathcal{U}$ over the unit cube $B = [0,1]^d$ that satisfy the following types of smoothness assumptions.

**Generalized Bandlimited functions.** Bandlimited functions are functions whose spectrum, or the set of frequencies that make up the function, is limited to a certain band or range of frequencies. This property makes bandlimited functions well-suited for certain applications, such as signal processing, where it is important to ensure that the signal does not contain frequencies outside of a certain range.

Generalized bandlimited functions are defined using a generalized analytic kernel $K : \mathbb{R} \to \mathbb{C}$, that generalizes the classic Fourier transform, and a band-limit $M \geq 1$. Formally, for a given function $f : \mathbb{R}^d \to \mathbb{R}$, we denote the set of all functions $F : [-M, M]^d \to \mathbb{C}$ that retrieve $f$ from the frequencies $\omega \in [-M, M]^d$ using the kernel $K$ as follows:

$$S_{f,K} := \left\{ F : [-M,M]^d \to \mathbb{R} \mid f(x) = \int_{[-M,M]^d} F(\omega) K(\omega \cdot x) \, \mathrm{d}\omega \right\}.$$

We define $F_f = \arg\min_{F \in S_{f,K}} \|F\|_{L^2([-M,M]^d)}$ to be the function in $S_{f,K}$ with the smallest $L_2$-norm. For instance, when $K(u) = \exp(iu)$, the function $F_f$ corresponds to the normalized standard Fourier transform of $f$, which is given by

$$F(\omega) = \tfrac{1}{(2\pi)^d} (\mathcal{F}f)(\omega) = \tfrac{1}{(2\pi)^d} \int_{\mathbb{R}^d} f(x) \exp(-i\omega \cdot x) \, \mathrm{d}x.$$

The space $\mathcal{H}_{K,M}(B)$ of generalized bandlimited functions is a Hilbert space of functions that can be represented as a weighted sum of the function $K$ over a finite domain. This space is equipped with an inner product and a norm, which allow us to measure the similarity and magnitude of these functions, respectively. We define $\mathcal{H}_{K,M}(B)$ as the functions $f : B \to \mathbb{R}$ such that

$$\mathcal{H}_{K,M}(B) := \left\{ \forall x \in B, f(x) = \int_{[-M,M]^d} F(\omega) K(\omega \cdot x) \, \mathrm{d}\omega \mid F \in L^2([-M,M]^d) \right\}.$$

The inner product and norm in this space are defined as follows: $\langle f, g \rangle_{\mathcal{H}_{K,M}(B)} = \int_{[-M,M]^d} F_f(\omega) \overline{F_g}(\omega) \, d\omega$, $\|f\|_{\mathcal{H}_{K,M}(B)} = \|F_f\|_{L^2([-M,M]^d)}$.

One of the key properties of generalized bandlimited functions is that they can be completely reconstructed from a discrete set of samples. This is known as the Shannon-Nyquist theorem (Shannon, 1949), and it is an important result in the field of signal processing and communication. An interesting consequence of this theorem is that even in high-dimensions, where seemingly unpredictable geometrical phenomena may occur (e.g. Blum et al. (2020), Chapter 2), a bandlimited function can still be perfectly reconstructed given its values at the Nyquist frequency. For more details, see Appendix A.

**Sobolev spaces.** Sobolev spaces are one of the most extensively studied classes of functions in approximation theory (DeVore & Lorentz, 1993; Yarotsky, 2017; Liang & Srikant, 2017). These spaces consist of functions with bounded or $p$-integrable distributional derivatives up to a certain order. As already discussed in the introduction, they are particularly useful in the study of PDEs.

We first define the $L_p$ norm, $1 \leq p < \infty$, of a given function $f : \Omega \to \mathbb{R}$ as $\|f\|_{L_p(\Omega)} = (\int_\Omega |f(x)|^p \, dx)^{1/p}$, where $\Omega \subset \mathbb{R}^d$ is measurable. When $p = \infty$, the essential supremum norm is used. A function $f$ is said to be in $L_p(\Omega)$ if $\|f\|_{L_p(\Omega)} < \infty$.

Let $r \in \mathbb{N}$ and $1 \leq p < \infty$. The Sobolev space $\mathcal{W}^{r,p}(B)$ consists of functions $f : B \to \mathbb{R}$ with $r$-distributional derivatives in $L_p$. The Sobolev norm $\|\cdot\|_{\mathcal{W}^{r,p}(B)}$ is defined as

$$\|f\|_{\mathcal{W}^{r,p}(B)} = \sum_{\alpha : |\alpha|_1 \leq r} \|D^\alpha f\|_{L_p(B)},$$

where $\alpha = (\alpha_1, \ldots, \alpha_d) \in \{0, \ldots, r\}^d$, $|\alpha|_1 = \alpha_1 + \cdots + \alpha_d$, and $D^\alpha f$ is the respective distributional derivative. We also present the semi-norm:

$$|f|_{r,p} := \sum_{\alpha : |\alpha|=r} \|D^\alpha f\|_{L_p(B)}.$$

Classic results in the literature show that the number of parameters needed to approximate functions in $\mathcal{W}^{r,\infty}$ up to error $\epsilon$ is lower bounded by $\Omega(\epsilon^{-d/r})$ (DeVore et al., 1989). This exponential dependence on $d$ is known as the "curse of dimensionality".

**Sobolev-Type Balls.** Sobolev-Type balls are typically subsets of Sobolev spaces with additional stronger properties (Barron, 1993; Jones, 1992; Pinkus, 1985; Wahba, 1990; Blanchard & Bennouna, 2022). One such property is that the magnitude of the function's Fourier transform, $|\mathcal{F}f(\omega)|$, decays fast enough as $|\omega|$ approaches infinity. These constraints are imposed by comparing the magnitude of the Fourier transform of the function, $\mathcal{F}f(\omega)$, with the magnitude of $|\omega|^r$ for some number $r > 0$. In this paper we define a generalized form of Sobolev-type balls:

$$\mathcal{B}_{r,\rho} = \left\{ f : \mathbb{R}^d \to \mathbb{R}, f \in L_2(\mathbb{R}^d) : \frac{1}{(2\pi)^d} \int_{\mathbb{R}^d} |(\mathcal{F}f)(\omega)| \, d\omega \leq 1, \frac{1}{(2\pi)^d} \int_{\mathbb{R}^d} |\omega|^r |(\mathcal{F}f)(\omega)|^\rho \, d\omega \leq 1 \right\}.$$

In (Barron, 1993), they explored the ability of neural networks to approximate functions in the space $P_1 = \{ f \in L_2(\mathbb{R}^d) \mid \int_{\mathbb{R}^d} |\omega| |(\mathcal{F}f)(\omega)| < \infty \}$. We now demonstrate that any function in $P_1$ has a normalized representation in $\mathcal{B}_{1,1}$. For this, we show that the conditions on $P_1$ allow us to bound $\|\mathcal{F}f\|_{L_1(\mathbb{R}^d)}$. Namely,

$$
\begin{aligned}
\int_{\mathbb{R}^d} |(\mathcal{F}f)(\omega)| \, d\omega &= \int_{[-1,1]^d} |(\mathcal{F}f)(\omega)| \, d\omega + \int_{\mathbb{R}^d \setminus [-1,1]^d} |(\mathcal{F}f)(\omega)| \, d\omega \\
&\leq C_1 \left( \int_{[-1,1]^d} |(\mathcal{F}f)(\omega)|^2 \, d\omega \right)^{1/2} + \int_{\mathbb{R}^d \setminus [-1,1]^d} |\omega| |(\mathcal{F}f)(\omega)| \, d\omega \\
&\leq C_2 \|f\|_2 + \int_{\mathbb{R}^d} |\omega| |(\mathcal{F}f)(\omega)| \, d\omega.
\end{aligned}
$$

Therefore, any function in the space $P_1$ can be scaled to a function in $\mathcal{B}_{1,1}$. Specifically, they showed that sigmoidal neural networks with a bounded depth and $\mathcal{O}(d\epsilon^{-2})$ neurons can approximate such functions with error at most $\epsilon$.

In this paper we will focus on the space $\mathcal{B}_{2r,2}$. In Lemma 2 (Appendix D) we show that this space is embedded as a proper subset of $\mathcal{W}^{r,2}$, with the following norm:

$$\|f\|_{\mathcal{B}_{2r,2}} \;=\; \left( \frac{1}{(2\pi)^d} \int_{\mathbb{R}^d} (1 + |\omega|^{2r}) |(\mathcal{F}f)(\omega)|^2 \; d\omega \right)^{1/2}. \tag{1}$$

In Lemma 2 (Appendix D), we also show an example of why the Sobolev-type ball $\mathcal{B}_{2r,2}$ is of particular interest. Specifically, we show that the additional condition that $\mathcal{F}f(\omega) \in L_1$ defines a subspace of $\mathcal{W}^{r,2}$ that can be easily approximated using multiplicative networks. It is worth mentioning that Pinkus (1985) showed that using traditional basis functions, functions of the space $P_2 = \{f \in L_2(\mathbb{R}^d) \mid \int_{\mathbb{R}^d} |\omega|^{2r} |(\mathcal{F}f)(\omega)|^2 < \infty\}$ may be approximated with error at most $\epsilon$ using $\mathcal{O}(\epsilon^{-d/2r})$ parameters. In this paper, we show that by enforcing an additional constraint on the $L_1$ norm of the Fourier transform, we can circumvent exponential dependence on the dimension $d$.

## 2.2 Neural Network Architectures

In the previous section, we described a setting in which a class of candidate functions $\mathcal{H}$ serve as approximators to a class of target functions $\mathcal{U}$. In this work, we compare the approximation guarantees of standard multilayer perceptrons and a generic set of neural networks that incorporate multiplication operations. Our goal is to understand whether multiplication layers can provide better guarantees for generalized bandlimited functions and $\mathcal{B}_{2r,2}$.

**Multilayer perceptrons.** A multilayer perceptron is a neural network architecture that consists of $L$ layers of affine linear transformations composed with element-wise non-linear activation functions (e.g., the ReLU function). Typically, the last layer does not include a non-linear activation.

**Definition 1** (Multilayer perceptron). *A multilayer perceptron $f = y_{L,1} : \mathbb{R}^{p_0} \to \mathbb{R}$ is defined by a set of functions $\bigcup_{i=0}^{L} \{y_{i,j}\}_{j=1}^{p_i}$. Each function $y_{i,j} : \mathbb{R}^{p_0} \to \mathbb{R}$ (also known as a neuron) is recursively computed in the following manner*

$$\begin{aligned} y_{L,j}(x) &= \langle w_{L,j}, y_{L-1}(x) \rangle + b_{L,j} \\ y_{i,j}(x) &= \sigma(\langle w_{i,j}, y_{i-1}(x) \rangle + b_{i,j}) \\ y_{0,j}(x) &= x_j, \end{aligned}$$

*where $i \in [L-1]$, $j \in [p_i]$, and $w_{i,j} \in \mathbb{R}^{p_{i-1}}$ and $b_{i,j} \in \mathbb{R}$ are the weights and a bias of the neuron $y_{i,j}$ and $y_i = (y_{i,1}, \ldots, y_{i,p_i})$. The function $\sigma : \mathbb{R} \to \mathbb{R}$ is a non-linear activation function.*

In this work, we focus on neural networks with ReLU activations, which are defined as $\sigma(x) = \max(0, x)$. However, it is worth noting that other activation functions have been proposed in the literature, such as sigmoidal functions that are measurable functions $\eta$ that satisfy $\eta(x) \to 0$ as $x \to -\infty$ and $\eta(x) \to 1$ as $x \to \infty$.

**Multiplicative neural networks.** In this work, we are interested in comparing the approximation abilities of standard ReLU networks, with that of neural networks that incorporate multiplication layers (also known as product units (Durbin & Rumelhart, 1989)). In order to fully understand the added benefits of multiplication gates, we ask the following question: *Are multiplications between neurons sufficient to substitute the non-linear activations in multilayer perceptrons?*

**Definition 2** (Multiplicative network). *A multiplicative neural network is a function $f = y_{L,1} : \mathbb{R}^{p_0} \to \mathbb{R}$ defined by a set of univariate functions $\bigcup_{i=1}^{L} \{y_{i,j}\}_{j=1}^{p_i}$. For each neuron, $y_{i,j} : \mathbb{R}^{p_0} \to \mathbb{R}$ there exists a pair of indices $j_1, j_2 \in \{1, \ldots, p_{i-1}\}$ (potentially dependent on $i, j$) such that $y_{i,j}$ is defined as follows*

$$\begin{aligned} y_{i,j}(x) &= \langle w_{i,j}, y_{i-1}(x) \rangle + a_{i,j} y_{i-1,j_1}(x) y_{i-1,j_2}(x) + b_{i,j} \\ y_{0,j}(x) &= x_j, \end{aligned}$$

*where $a_{i,j}, b_{i,j} \in \mathbb{R}$, $w_{i,j} \in \mathbb{R}^{p_{i-1}}$ are trainable parameters, and $y_i = (y_{i,1}, \ldots, y_{i,p_i})$.*

Multiplicative networks differ from multilayer perceptrons as they rely on (non-linear) multiplicative gates between neurons instead of element-wise activation functions. Each neuron $y_{i,j}$ is very similar to a standard neuron in a multilayer perceptron, with the distinction that a weighted multiplication between a single pair of neurons $y_{i-1,j_1}, y_{i-1,j_2}$ is added to the output. The total number of trainable parameters in a multiplicative layer is $p_{i+1}p_i + 2p_i$, which is comparable to the $p_{i+1}p_i + p_i$ in a fully-connected layer. Following Yarotsky (2017); Blanchard & Bennouna (2022) we measure the complexity of a neural network using its depth $L$ and its total number of neurons $G = \sum_{i=1}^{L-1} p_i$.

**Self-Attention and multiplicative layers.** Let us describe a single-headed self-attention operation in the original Transformer (Vaswani et al., 2017). Each layer $i \in [L]$ of a depth-$L$ Transformer encoder is defined as follows. The input to the $i$th layer is a sequence of $N$ tokens, denoted by $x_i = \{x_{i,j}\}_{j=1}^N$, where each $x_{i,j} \in \mathbb{R}^{d_x}$ represents the $j$th token of the $i$th layer. To compute the output of the $i$th layer at a particular position $e \in [N]$, we use the following formula:

$$f_{\mathrm{SA}}^{i,e}(x_i) \;=\; \sum_{j=1}^N \mathrm{softmax}_j \left( \tfrac{1}{\sqrt{d_a}} \langle W^{Q,i} x_{i,e}, W^{K,i} x_{i,e} \rangle \right) W^{V,i} x_{i,j},$$

where $\mathrm{softmax}_j(f(x)) = \exp(f(x)_j)/\sum_{j'} \exp(f(x)_{j'})$ is the softmax operator, and the trainable weight matrices $W^{K,i}$, $W^{Q,i}$, $W^{V,i} \in \mathbb{R}^{d_a \times d_x}$ convert the representation from its dimension $d_x$ into the attention dimension $d_a = d_x$, creating 'Key', 'Query', and 'Value' representations, resp. As can be seen, the self-attention layers use multiplicative connections when computing the following inner product $\langle W^{Q,i} x_{i,e}, W^{K,i} x_{i,e} \rangle$. This operation computes multiplications between the coordinates of transformations of the same token $x_{i,e}$. In other words, it can be thought of as computing a multiplicative layer on an input vector $x_{i,e}$. As a side note, in addition to self-attention layers, transformers also incorporate commonly used layers such as fully-connected layers, residual connections, and normalization layers, which are not the focus of this paper.

## 3   Representation Power of Multiplicative Neural Networks

In this section, we explore the expressive power of neural networks with multiplication layers. We first demonstrate that these networks can easily represent polynomial functions, which we then use to approximate bandlimited functions. This allows us to approximate functions in the space $\mathcal{B}_{2r,2}$ without suffering from the curse of dimensionality. Specifically, we prove the following lemma:

**Lemma 1.** *For any polynomial $p_n : \mathbb{R} \to \mathbb{R}$ of degree $n$ of the form $p_n(x) = \sum_{k=0}^n c_k x^k$, there exists a multiplicative neural network $f_n^{\mathrm{POL}} : \mathbb{R}^3 \to \mathbb{R}$, of depth $L_n = \mathcal{O}(n)$ with $G_n = \mathcal{O}(n)$ neurons that satisfies $f_n^{\mathrm{POL}}(x, x, c_0) = p_n(x)$ for $c_0 \in \mathbb{R}$, and $x \in \mathbb{R}$.*

Next, we may leverage the above lemma and proceed to show how multiplicative networks can be used to approximate analytic functions For this purpose, we recall the notion of the Bernstein s-ellipse, which is a geometric shape defined on the complex plane that is useful in approximation theory.

**Definition 3.** *Let $M > 0$, $s > 1$ be two scalars. The Bernstein s-ellipse on $[-M, M]$ is defined as follows*

$$E_s^M \;=\; \left\{ x + iy \in \mathbb{C} : \tfrac{x^2}{(a_s^M)^2} + \tfrac{y^2}{(b_s^M)^2} = 1 \right\},$$

*whose semi-major and semi-minor axes are $a_s^M = M\frac{s+s^{-1}}{2}$ and $b_s^M = M\frac{s-s^{-1}}{2}$.*

The parameter $s$ controls the shape of the ellipse. As $s$ increases, the ellipse converges to a circle. Before stating our result in Theorem 2, we recall the following theorem of Trefethen (2019):

**Theorem 1** (Theorem 8.2 of Trefethen (2019))**.** *Let $M > 0, s > 2$ be scalars and $K : [-M, M] \to \mathbb{R}$ be an analytic function that is analytically continuable to the ellipse $E_s^M$, where it satisfies $\sup_{x \in E_s^M} |K(x)| \le C_K$ for some constant $C_K > 0$. For every $n \in \mathbb{N}$, there exists a polynomial $h_n : \mathbb{R} \to \mathbb{C}$ of degree $n$, such that,*

$$\|h_n - K\|_{L^\infty([-M,M])} \;\le\; \frac{2 C_K s^{-n}}{s-1}.$$

Theorem 1 states that any analytic function $K$ that is bounded on the Bernstein s-ellipse $E_s^M$ can be approximated by a polynomial of degree $n$, with the error decreasing exponentially as the degree $n$ increases.

As we show next, the use of multiplication layers in neural networks can improve the efficiency of function approximation in certain cases. The following theorem shows that deep multiplicative networks can approximate real-valued analytic functions on bounded intervals.

**Theorem 2.** *Let $M \geq 1$, $s > 2$, $C_K > 0$ and $\epsilon \in (0,1)$ be scalars. Then, for any real-valued analytic function $K : [-M,M] \to \mathbb{R}$ that is analytically continuable to the ellipse $E_s^M$ where $|K(x)| \leq C_K$, there exists a deep multiplicative network $f^{\mathrm{MA}} : [-M,M]^3 \to \mathbb{R}$ (MA stands for 'Multiplicative Analytic') of depth $L_\epsilon = \mathcal{O}(\frac{1}{\log_2 s} \log_2 \frac{C_K}{\epsilon})$ with $G_\epsilon = \mathcal{O}(\frac{1}{\log_2 s} \log_2 \frac{C_K}{\epsilon})$ neurons that satisfies $\|f^{\mathrm{MA}}(x,x,x) - K(x)\|_{L^\infty([-M,M])} \leq \epsilon$.*

Theorem 2 establishes that deep multiplicative networks with second-degree multiplications can approximate any real-valued analytic function that is bounded on the Bernstein s-ellipse $E_s^M$ with error bounded by a quantity that decreases exponentially with the depth of the network.

In (Montanelli & Du, 2021), the authors show that the kernel may be approximated using ReLU networks with depth $L_\epsilon = \mathcal{O}(\frac{1}{\log_2^2 s} \log_2^2 \frac{C_K}{\epsilon})$ with $G_\epsilon = \mathcal{O}(\frac{1}{\log_2^2 s} \log_2^2 \frac{C_K}{\epsilon})$ neurons. Thus, our approach achieves a $\mathcal{O}(\log(1/\epsilon))$ improvement in depth and a $\mathcal{O}(\log(1/\epsilon))$ in the number of neurons.

**Approximation of bandlimited functions.** As a next step, we study the ability of neural networks to approximate bandlimited functions. The following theorem shows that a deep multiplicative network can approximate in $B = [0,1]^d$, a bandlimited function up to a given error tolerance using a relatively small number of neurons and depth.

**Theorem 3.** *Let $\epsilon \in (0,1)$, $M > 1, d \geq 2$, and $K : \mathbb{R} \to \mathbb{C}$ be an analytic kernel that holds the assumptions of Theorem 2 with respect to $s > 2, C_K > 0$, and bounded by a constant $D_K \in (0,1]$ on $[-dM, dM]$. Let $f$ be a real-valued function in $\mathcal{H}_{K,M}(B)$. Further, let $F : [-M,M]^d \to \mathbb{C}$ be a square-integrable function such that $f(x) = \int_{[-M,M]^d} F(\omega) K(\omega \cdot x) \, d\omega$. We define $C_F = \int_{\mathbb{R}^d} |F(\omega)| \, d\omega = \int_{[-M,M]^d} |F(\omega)| \, d\omega$. Then, there exists a deep multiplicative network $f^{\mathrm{MBL}} : B \to \mathbb{R}$ (MBL stands for 'Multiplicative bandlimited') of depth $L_\epsilon = \mathcal{O}\left(\frac{1}{\log_2 s} \log_2 \frac{C_F C_K}{\epsilon}\right)$ with $G_\epsilon = \mathcal{O}\left(\frac{C_F^2}{\epsilon^2 \log_2 s} \log_2 \frac{C_F C_K}{\epsilon}\right)$ neurons that satisfies $\|f^{\mathrm{MBL}} - f\|_{L^2(B)} \leq \epsilon$.*

According to the above theorem, one can approximate bandlimited functions up to error $\epsilon$ using multiplicative neural networks of depth $L_\epsilon = \mathcal{O}(\log(1/\epsilon))$ using $G_\epsilon = \mathcal{O}(\epsilon^{-2} \log(1/\epsilon))$ neurons. In comparison, Montanelli & Du (2021) showed that one can approximate bandlimited functions to the same level of approximation using ReLU networks of depths $L_\epsilon = \mathcal{O}(\log^2(1/\epsilon))$ with $G_\epsilon = \mathcal{O}(\epsilon^{-2} \log^2(1/\epsilon))$ neurons. This result demonstrates the parameter efficiency of multiplicative networks in comparison with standard ReLU networks.

**Approximation of Sobolev Functions.** We now turn to show results for Sobolev-Type functions. We use the results on bandlimited functions shown in Theorem 3 to approximate functions in $\mathcal{B}_{2r,2}$. We show that using slightly stronger assumptions, we get an approximation rate comparable to those shown by Barron (1993) (where the network is a shallow sigmoidal network model) using multiplicative networks (i.e. without non-linear activations).

**Theorem 4.** *Let $d \geq 2, r \in \mathbb{N}$, $f \in \mathcal{B}_{2r,2}$ and $\epsilon > 0$. There exists a deep ReLU network $f^{\mathrm{RS}}$ (standing for "ReLU Sobolev") with a depth of $L_\epsilon = \mathcal{O}(d^2 \epsilon^{-2/r})$ and $G_\epsilon = \mathcal{O}(d^2 \epsilon^{-(2+2/r)})$ neurons, such that $\|f^{\mathrm{RS}} - f\|_{L_2(B)} \leq \epsilon$.*

**Theorem 5.** *Let $d \geq 2, r \in \mathbb{N}$, $f \in \mathcal{B}_{2r,2}$ and $\epsilon > 0$. There exists a deep ReLU network $f^{\mathrm{MS}}$ (standing for "Multiplicative Sobolev") with a depth of $L_\epsilon = \mathcal{O}(d \epsilon^{-1/r})$ and $G_\epsilon = \mathcal{O}(d \epsilon^{-(2+(1/r))})$ neurons, such that $\|f^{\mathrm{MS}} - f\|_{L_2(B)} \leq \epsilon$.*

In the following corollary, we demonstrate an application of Theorem 5 for approximating functions in the Sobolev space $\mathcal{W}^{r,2}$ with an integrable Fourier transform.

**Corollary 1.** *Let $d \geq 2, r \in \mathbb{N}$, $f \in \mathcal{W}^{r,2}$ such that $\|\mathcal{F}f\|_{L_1(\mathbb{R}^d)} < \infty$, and $\epsilon > 0$. There exists a deep ReLU network $f^{\mathrm{MS}}$ (standing for "Multiplicative Sobolev") with a depth of $L_\epsilon = \mathcal{O}(\epsilon^{-1/r})$ and $G_\epsilon = \mathcal{O}(\epsilon^{-(2+(1/r))})$ neurons, such that $\|f^{\mathrm{MS}} - f\|_{L_2(B)} \leq C \max\{|f|_{r,2}^2, \|\mathcal{F}f\|_1\}\epsilon$, where $C > 0$ is a constant dependent on $r, d$.*

*Proof of Theorem 5.* Let $f \in \mathcal{B}_{2r,2}$. We would like to approximate $f$ using a bandlimited function $f_M$ and then approximate $f_M$ using a multiplicative neural network $f^{\text{MS}}$ (standing for "Multiplicative Sobolev"). Let $M > 1$ be a band.

We recall the Inverse Fourier transform given by $(\mathcal{F}^{-1}g)(x) = \frac{1}{(2\pi)^d} \int_{\mathbb{R}^d} g(\omega) \exp(i\omega \cdot x) \, d\omega$. We define the bandlimiting of $f : \mathbb{R}^d \to \mathbb{R}$ as $f_M = \mathcal{F}^{-1}(\mathcal{F}f \mathbb{1}_{[-M,M]^d})$, such that $f_M \in \mathcal{H}_{K,M}(B)$ for $K(u) = \exp(iu)$ and $F = \frac{1}{(2\pi)^d} \mathcal{F}f$. We have

$$\|f - f_M\|_{L_2(B)} \leq \left( \frac{1}{(2\pi)^d} \left\| \mathcal{F}f - \mathcal{F}f \mathbb{1}_{[-M,M]^d} \right\|_{L_2(\mathbb{R}^d)}^2 \right)^{1/2} = \left( \frac{1}{(2\pi)^d} \int_{\mathbb{R}^d \setminus [-M,M]^d} |\mathcal{F}f(\omega)|^2 \, d\omega \right)^{1/2}.$$

For any $\omega \in \mathbb{R}^d \setminus [-M,M]^d$, we have $|M^{-1}\omega|^{2r} \geq 1$. Therefore, since $f \in \mathcal{B}_{2r,2}$

$$\left( \frac{1}{(2\pi)^d} \int_{\mathbb{R}^d \setminus [-M,M]^d} |\mathcal{F}f(\omega)|^2 \, d\omega \right)^{1/2} \leq \left( \frac{1}{(2\pi)^d} \int_{\mathbb{R}^d \setminus [-M,M]^d} \left| M^{-1}\omega \right|^{2r} |\mathcal{F}f(\omega)|^2 \, d\omega \right)^{1/2}$$

$$\leq M^{-r} \left( \frac{1}{(2\pi)^d} \int_{\mathbb{R}^d} |\omega|^{2r} |\mathcal{F}f(\omega)|^2 \, d\omega \right)^{1/2}$$

$$\leq M^{-r}.$$

For any $\epsilon > 0$, we set $M = (2/\epsilon)^{1/r}$. We will construct a neural network $f^{\text{MS}}$ to approximate the bandlimited function $f_M$, such that $\|f_M - f^{\text{MS}}\|_{L_2(B)} \leq \epsilon/2$. Assuming we have constructed such $f^{\text{MS}}$, then by the triangle inequality we then arrive at

$$\|f - f^{\text{MS}}\|_{L_2(B)} \leq \|f - f_M\|_{L_2(B)} + \|f_M - f^{\text{MS}}\|_{L_2(B)} \leq M^{-r} + \epsilon/2 \leq \epsilon.$$

Let us define the function $F : [-M,M]^d \to \mathbb{C}$ as follows $F(\omega) = \frac{1}{(2\pi)^d}(\mathcal{F}f)(\omega)$. We then have the following identity

$$f_M(x) = \int_{[-M,M]^d} F(\omega)K(\omega \cdot x) \, d\omega = \int_{[-M,M]^d} \frac{1}{(2\pi)^d} \mathcal{F}f(\omega) \exp(i\omega \cdot x) \, d\omega.$$

We may now work under the conditions of Theorem 3. We consider that $C_F = \int_{[-M,M]^d} |F(\omega)| \, d\omega = \frac{1}{(2\pi)^d} \int_{[-M,M]^d} |\mathcal{F}f(\omega)| \, d\omega \leq 1$, where the inequality is due to the definition of $\mathcal{B}_{2r,2}$. The kernel $K(u) = \exp(iu)$ takes as input $u = \omega \cdot x$, for $x \in B$ and $\omega \in [-M,M]^d$. Therefore, $u \in [-dM, dM]$, and $K : [-dM, dM] \to \mathbb{R}$. Since $K$ is an entire function, it is continuable to $E_s^{dM}$ for any $s > 2$, let us choose $s = 4$. We notice that $a_4^{dM} := dM(4 + 4^{-1})/2$ is the larger axis, and therefore the maximal norm of $K$ on $E_s^{dM}$ is given by

$$C_K \leq \exp(dM \tfrac{4+4^{-1}}{2}).$$

Further, for any $u \in \mathbb{R}$ we have $|K(u)| \leq 1 = D_K$. We now approximate $f_M$ with a multiplicative network. Using the results of Theorem 3, there exists a deep multiplicative neural network $f^{\text{MS}}$ that approximates the bandlimited function $f_M$ in $L_2(B)$ with error bounded by $\epsilon/2$ and depth

$$
\begin{aligned}
L_\epsilon &\leq C_1 \frac{1}{\log_2 4} \log_2 \left( \frac{2 C_K C_F}{\epsilon} \right) \\
&\leq C_2 \frac{1}{\log_2 4} \log_2 \left( \frac{\exp(dM \frac{4+4^{-1}}{2})}{\epsilon} \right) \\
&= \frac{C_2}{2} \left( dM \frac{4 + 4^{-1}}{2 \log(2)} + \log_2(1/\epsilon) \right) \\
&\leq \frac{C_2}{2} (4dM + \log_2(1/\epsilon)) \\
&\leq C_2 \left( 4d\epsilon^{-1/r} + \log_2(1/\epsilon) \right) \\
&\leq C_3 \cdot d\epsilon^{-1/r},
\end{aligned}
\tag{2}
$$

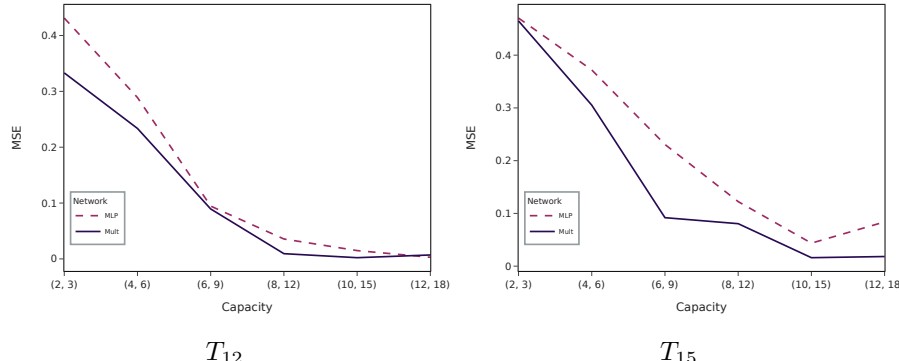

$T_{12}$                                              $T_{15}$

Figure 1: **Approximating Chebychev polynomials with networks of increasing complexity.** We present the minimal test mean squared error (MSE) loss for approximating Chebychev polynomials $T_n$ using networks with increasing complexity, utilizing 10 initialization seeds. The network complexity (x-axis) is defined as (depth, width) and is increased linearly.

for some constants $C_1, C_2, C_3 > 0$. The third step follows from $\log_2(xy) = \log_2(x) + \log_2(y)$ and $\log_a(\exp(x)) = x/\log(a)$ (for all $x, y, a > 0$) and the fifth inequality follows from substituting $M = (2/\epsilon)^{1/r}$. In addition, the number of neurons can be bounded by

$$
\begin{aligned}
G_\epsilon &\leq C_1' C_F^2 \cdot \epsilon^{-2} \log_2 \frac{2C_K C_F}{\epsilon} \\
&\leq C_2' \cdot \epsilon^{-2} \log_2 \frac{\exp(dM^{\frac{4+4^{-1}}{2}})}{\epsilon} \\
&\leq C_2' \cdot \epsilon^{-2}(3d(2/\epsilon)^{1/r} + \log_2(1/\epsilon)) \\
&\leq C_3' \cdot d\epsilon^{-(2+1/r)},
\end{aligned}
\tag{3}
$$

for some constants $C_1', C_2', C_3' > 0$.          □

Theorems 4-5 provide insights into several interesting properties of the Sobolev-Type ball $\mathcal{B}_{2r,2}$. First, we see that approximating these functions does not suffer from the curse of dimensionality that occurs when approximating the full Sobolev space. Secondly, for both ReLU networks and multiplicative networks, the bound decreases as the smoothness index $r$ increases. In fact, as $r$ approaches infinity, the bound approaches the one proposed in (Barron, 1993). Lastly, observe that for the same error tolerance $\epsilon$, multiplicative neural networks can approximate a target function $f \in \mathcal{B}_{2r,2}$ with a depth of $\mathcal{O}(d\epsilon^{-1/r})$ and $\mathcal{O}(d\epsilon^{-(2+1/r)})$ neurons, while standard ReLU neural networks require a depth of $\mathcal{O}(d^2\epsilon^{-2/r})$ and $\mathcal{O}(d^2\epsilon^{-(2+2/r)})$ neurons. This result demonstrates that multiplicative neural networks have stronger approximation guarantees when approximating functions in the Sobolev-Type space.

## 4 Experiments

In previous sections, we demonstrated that multiplicative networks have lower complexity requirements for approximating certain target functions when compared to multilayer perceptrons. To validate these results empirically, we analyzed the influence of network complexity such as depth and width on the network's approximation error. To do so, we approximated Chebyshev polynomials of varying degrees $T_n$ using multilayer perceptrons and multiplicative networks with increasing complexities (i.e., depth, width). As the degree of the Chebyshev polynomials increased, we expected multilayer perceptrons to struggle in approximating high-degree polynomials. Conversely, as proven in Lemma 1, $T_n$ can be perfectly constructed using a multiplicative network with depth $L_n = n$ and width $G_n = 3n$.

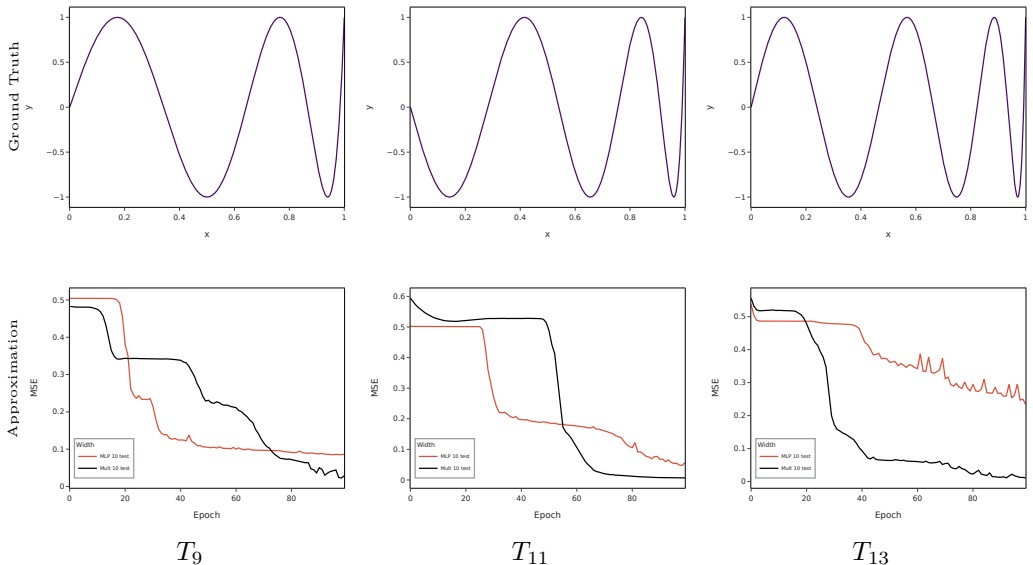

Figure 2: **Approximating Chebychev polynomials.** We present the minimal test mean squared error (MSE) loss for approximating Chebychev polynomials $T_n$, using networks of depth $n$ and width 10, with 10 initialization seeds. The first row displays the actual polynomials, and the second row shows the approximation errors using both multilayer perceptrons and multiplicative networks.

To validate Theorems 4 and 5, we approximated randomly generated Gaussian mixtures. Specifically, we consider functions of the following form

$$f(x; \mu, \Sigma) := \frac{1}{N} \sum_{i=1}^{N} \frac{1}{(2\pi)^{d/2} |\Sigma_i|^{1/2}} \exp\left(-\frac{1}{2}(x - \mu_i)^\top \Sigma_i^{-1}(x - \mu_i)\right),$$

where $\Sigma \in \mathbb{R}^{N \times d \times d}$ and $\mu \in \mathbb{R}^{N \times d}$. We sample $\mu_i$ uniformly from $[0,1]^d$ and $\Sigma_i$ uniformly from $[0, 0.4]^d$. It is straightforward to show that Gaussian mixtures are in the Schwartz class (Schwartz, 1950). This implies that they are in $\mathcal{W}^{r,2}$, and that $\mathcal{F}f \in L_1(\mathbb{R}^d)$, which in turn means that they are also members of $\mathcal{B}_{2r,2}$ up to a constant factor (similar to the target functions considered in Corollary 1).

Estimating the approximation error of a model for a given target function is generally a challenging problem. In the literature, this is usually done by training the model to fit the target function. However, the approximation error may differ from the estimation due to optimization issues, such as sub-optimal initialization and improperly tuned hyperparameters. To obtain a more precise estimation of the approximation error, we trained each model using 10 different initialization seeds and selected the minimal test mean squared error (MSE) loss of the model. This approach helped us avoid problems related to poor initialization.

In Figure 1, we approximated $T_{12}$ and $T_{15}$ using networks whose depth and width varied simultaneously. The models were trained using the Adam optimizer with a learning rate of 0.001 for 100 epochs and a batch size of 64 to minimize their Mean Squared Error (MSE) (with respect to the target function). We used 1000 training and 1000 test samples that were uniformly selected at random from $[0, 1]$. As shown, multiplicative networks achieved better results (lower errors) compared to multilayer perceptrons with the same depth and width.

In Figure 2, we approximated $T_n$ of degrees $n = 9, 11, 13$ using multilayer perceptrons and multiplicative networks with depth $n$ and width 10. We used the same training and testing processes as in the previous experiment. As can be seen, multiplicative networks converged to a zero loss faster and provided better approximation rates for the given polynomials. For the code for generating the experiments, see Appendix B.

We conducted experiments to approximate the mixtures of Gaussians using multilayer perceptrons and multiplicative networks, while varying the dimensions and number of Gaussians ($d$ and $N$, respectively). All

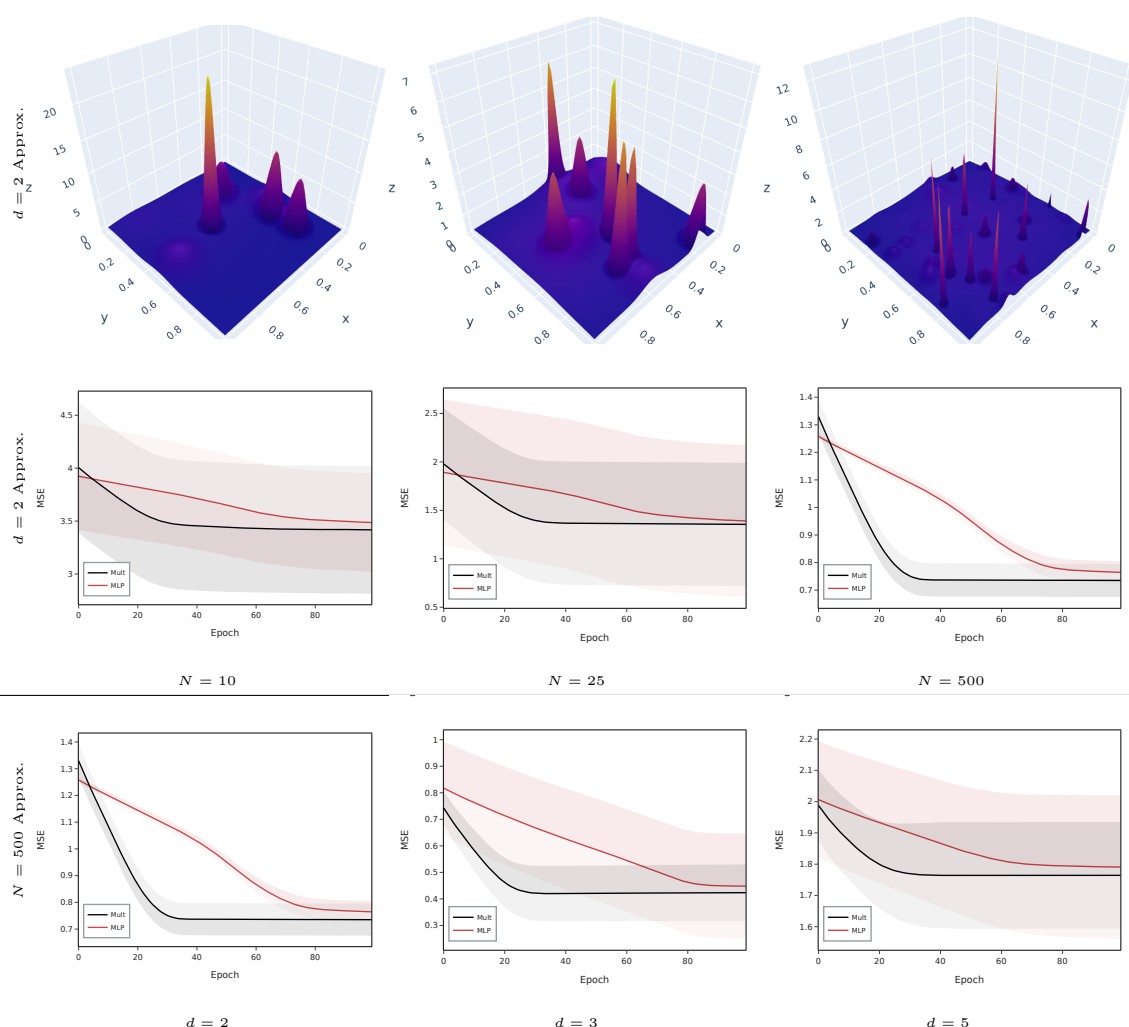

Figure 3: **Approximating Gaussian mixtures with a varying number of Gaussians and dimensions.** We plot the test mean squared error (MSE) loss obtained by approximating Gaussian mixtures $f(x; \mu, \Sigma)$ where $\mu \in \mathbb{R}^{N \times d}$ and $\Sigma \in \mathbb{R}^{N \times d \times d}$ using multiplicative and ReLU networks for different values of $d$ and $N$. We report their errors averaged over 20 initialization seeds along with their corresponding 95% confidence intervals.

networks had a depth of 5 and a width of 10. We plotted the averaged approximation errors during training over 20 initialization seeds, along with their corresponding 95% confidence intervals, while fixing the target function. The training was performed with the same hyperparameters as in the previous experiments, except for the learning rate, which was set to $10^{-4}$.

Our results, shown in Figure 3, indicate that multiplicative networks outperform multilayer perceptrons in terms of approximation errors across various dimensions and numbers of Gaussians. Interestingly, we also observed that training with multiplicative networks converges much faster than with multilayer perceptrons.

## 5 Conclusions

Previous papers have studied the approximation guarantees of standard fully-connected neural networks to approximate functions in the Barron space $\mathcal{B}_{1,1}$ (Barron, 1993), the space of bandlimited functions (Montanelli & Du, 2021), and the Korobov space (Blanchard & Bennouna, 2022). These studies have shown that fully-

connected networks can approximate a wide range of smooth functions without suffering from the curse of dimensionality and have provided insights into the tradeoffs between the width and depth of neural networks in learning certain types of functions. However, these results are limited to variants of fully-connected networks and do not provide information about other types of architectures.

In this paper, we extend these results by exploring the approximation guarantees of both multiplicative neural networks and standard fully-connected networks to approximate bandlimited functions and members of the Sobolev-Type ball $\mathcal{B}_{2r,2}$. Our results show that multiplicative neural networks achieve stronger approximation guarantees compared to standard ReLU networks. In addition, we show that, unlike the Barron space and the space of bandlimited functions, $\mathcal{B}_{2r,2}$ is a subset of the Sobolev space $\mathcal{W}^{r,2}$. Therefore, our results demonstrate that it is possible to avoid the curse of dimensionality for wide subsets of the Sobolev space.

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

## A Examples

**Example 1.** *Let $f \in L_2(\mathbb{R}^d)$ such that $f$ is Bandlimited function with band $\pi$ (i.e., $\text{supp}(\mathcal{F}f) \subset [-\pi, \pi]^d$). Let $\phi(x) = \text{sinc}(x) = \sin(\pi x)/\pi x$, and $\phi_k = \phi(\cdot - k)$ for all $k \in \mathbb{Z}^d$. By the Shannon-Nyquist theorem (Shannon, 1949), we have:*

$$f(x) = \sum_{k \in \mathbb{Z}^d} \langle f, \phi_k \rangle \phi_k(x) = \sum_{k \in \mathbb{Z}^d} f(k)\phi(x - k).$$

*This means that every Bandlimited function can be completely determined using a discrete set of integer samples. This result is particularly surprising for high-dimensional functions (d is large) since the maximal distance between a point $x$ and a sampling point grows with d. For example, the vertex of the unit cube in $\mathbb{R}^d$ is of distance $\sqrt{d}/2$ away from its center. Despite this, we can still recover samples from the integer vertices, even when the distances scale as $\sqrt{d}$. This property is useful when recovering high-dimensional functions using neural networks. See Figure 4 for illustration.*

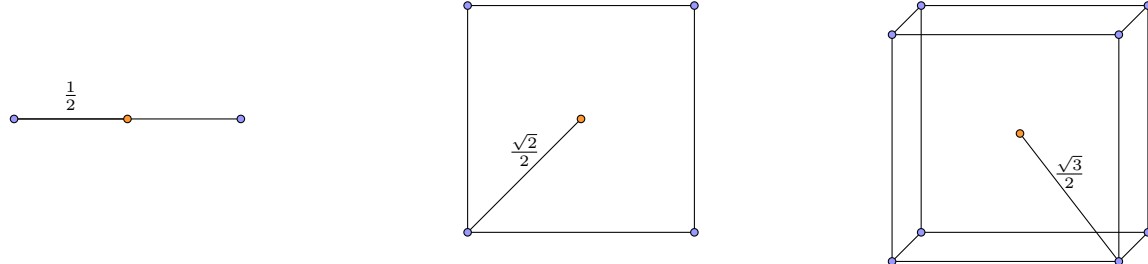

Figure 4: Illustration of Example 1. A bandlimited function $f$ can be reconstructed using a discrete set of values, despite the fact that the distances in each cube grow as $\mathcal{O}(\sqrt{d})$.

## B Code

In this section, we provide the code used to generate the experiments. The network architectures are described in Section 2.2. The "chebychev(n, x)" function calculates the Chebyshev polynomial of degree $n$ for a given input $x$. The "MLPPolynomialApproximation" and "MultPolynomialApproximation" classes are Pytorch Lightning (Falcon et al., 2019) implementations for the multilayer perceptrons and multiplicative networks, respectively. The "MultiplicativeLayer" class is based on the standard linear layer in PyTorch (Paszke et al., 2019). Since the neurons in each layer are interchangeable, we arbitrarily connected the first two neurons in each layer using a multiplicative operation. In each network, the layers are followed by ReLU activations.

```
from torch.nn.parameter import Parameter, UninitializedParameter
from torch.nn import init
from torch.nn import functional as F
import torch
from torch import Tensor
import torch.nn as nn
import torch.optim as optim
import pytorch_lightning as pl
from torch.utils.data import DataLoader, TensorDataset
from tqdm import tqdm
import numpy as np
from sklearn.model_selection import train_test_split

def chebychev(n, x):
    coefs = [0 for _ in range(n)] + [1]
```

```python
        return np.polynomial.chebyshev.Chebyshev(coefs)(x)

class MultiplicativeLayer(nn.Module):
    __constants__ = ['in_features', 'out_features']
    in_features: int
    out_features: int
    weight: Tensor

    def __init__(self, in_features: int=3, out_features: int=3, bias: bool = True,
                 device=None, dtype=None) -> None:
        factory_kwargs = {'device': device, 'dtype': dtype}
        super(AttentionLayer, self).__init__()
        self.in_features = in_features
        self.out_features = out_features

        self.weight = Parameter(torch.empty((out_features, in_features), **factory_kwargs)
            )
        self.bias = Parameter(torch.empty(out_features, **factory_kwargs))
        self.mult_weight = Parameter(torch.empty(1, **factory_kwargs))
        self.reset_parameters()

    def reset_parameters(self) -> None:
        init.kaiming_uniform_(self.weight, a=math.sqrt(5))
        if self.bias is not None:
            fan_in, _ = init._calculate_fan_in_and_fan_out(self.weight)
            bound = 1 / math.sqrt(fan_in) if fan_in > 0 else 0
            init.uniform_(self.bias, -bound, bound)
            init.uniform_(self.mult_weight, -bound, bound)

    def forward(self, x: Tensor) -> Tensor:
        bs, _ = x.shape
        x = F.linear(x, self.weight, self.bias)
        mult_output = torch.prod(x.gather(1, torch.tensor([0, 1]).repeat(bs, 1)), dim=1)
        mask = torch.zeros_like(x)
        mask[:, 1] = self.mult_weight*mult_output
        x = x + mask
        return x

class MultPolynomialApproximation(pl.LightningModule):
    def __init__(self, width=3, depth=3, poly_deg=3, use_relu=False):
        super().__init__()
        self.width = width
        self.depth = depth
        self.poly_deg = poly_deg
        self.layers = nn.ModuleList()
        self.layers.append(nn.Linear(1, width))
        self.train_loss = {}
        self.val_loss = {}
        for i in range(self.depth):
            self.layers.append(AttentionLayer(width, width))
        self.output = nn.Linear(width, 1)
        self.use_relu = use_relu

    def forward(self, x):
```

```python
        for layer in self.layers:
            x = layer(x)
            if self.use_relu:
                x = torch.relu(x)
        return self.output(x)

    def training_step(self, batch, batch_idx):
        x, y = batch
        y_hat = self.forward(x)
        loss = torch.mean((y_hat - y) ** 2)
        self.log("train_loss", loss)
        return {"loss": loss}

    def training_epoch_end(self, outputs) -> None:
        loss = sum(output['loss'] for output in outputs) / len(outputs)
        self.train_loss[self.current_epoch] = loss.item()

    def validation_step(self, batch, batch_idx):
        x, y = batch
        y_hat = self.forward(x)
        loss = torch.mean((y_hat - y) ** 2)
        self.log("val_loss", loss, prog_bar=True)
        return {"val_loss": loss}

    def validation_epoch_end(self, outputs) -> None:
        avg_loss = torch.stack([x["val_loss"] for x in outputs]).mean()
        self.val_loss[self.current_epoch] = avg_loss.item()

    def validation_end(self, outputs):
        avg_loss = torch.stack([x["val_loss"] for x in outputs]).mean()
        self.log("val_loss", avg_loss)
        self.val_loss[self.current_epoch] = avg_loss.item()
        return {"val_loss": avg_loss}

    def prepare_data(self, val_split=0.2):
        N = 1000
        X = torch.rand(N, 1)
        Y = chebychev(self.poly_deg, X[:, 0]).view(-1, 1)
        X_train, X_val, Y_train, Y_val = train_test_split(X, Y, test_size=val_split)
        self.train_dataset = torch.utils.data.TensorDataset(X_train, Y_train)
        self.val_dataset = torch.utils.data.TensorDataset(X_val, Y_val)

    def train_dataloader(self):
        return torch.utils.data.DataLoader(self.train_dataset, batch_size=64, shuffle=True
            )

    def val_dataloader(self):
        return torch.utils.data.DataLoader(self.val_dataset, batch_size=32, shuffle=False)

    def configure_optimizers(self):
        optimizer = optim.Adam(self.parameters(), lr=1e-3)
        return optimizer
```

```python
class MLPPolynomialApproximation(pl.LightningModule):
    def __init__(self, width=5, depth=3, poly_deg=3):
        super().__init__()
        self.width = width
        self.depth = depth
        self.poly_deg = poly_deg
        self.layers = nn.ModuleList()
        self.layers.append(nn.Linear(1, width))
        self.train_loss = {}
        self.val_loss = {}
        for i in range(self.depth):
            self.layers.append(nn.Linear(width, width))
        self.output = nn.Linear(width, 1)

    def forward(self, x):
        for layer in self.layers:
            x = torch.relu(layer(x))
        return self.output(x)

    def training_step(self, batch, batch_idx):
        x, y = batch
        y_hat = self.forward(x)
        loss = torch.mean((y_hat - y) ** 2)
        self.log("train_loss", loss)
        return {"loss": loss}

    def training_epoch_end(self, outputs) -> None:
        loss = sum(output['loss'] for output in outputs) / len(outputs)
        self.train_loss[self.current_epoch] = loss.item()

    def validation_step(self, batch, batch_idx):
        x, y = batch
        y_hat = self.forward(x)
        loss = torch.mean((y_hat - y) ** 2)
        self.log("val_loss", loss, prog_bar=True)
        return {"val_loss": loss}

    def validation_epoch_end(self, outputs) -> None:
        avg_loss = torch.stack([x["val_loss"] for x in outputs]).mean()
        self.val_loss[self.current_epoch] = avg_loss.item()

    def validation_end(self, outputs):
        avg_loss = torch.stack([x["val_loss"] for x in outputs]).mean()
        self.log("val_loss", avg_loss)
        self.val_loss[self.current_epoch] = avg_loss.item()
        return {"val_loss": avg_loss}

    def prepare_data(self, val_split=0.2):
        N = 1000
        X = torch.rand(N, 1)
        Y = chebychev(self.poly_deg, X[:, 0]).view(-1, 1)
        X_train, X_val, Y_train, Y_val = train_test_split(X, Y, test_size=val_split)
        self.train_dataset = torch.utils.data.TensorDataset(X_train, Y_train)
        self.val_dataset = torch.utils.data.TensorDataset(X_val, Y_val)
```

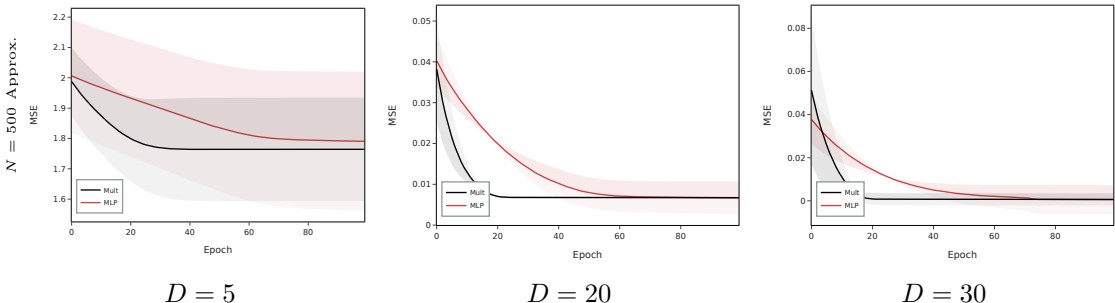

Figure 5: **Approximating Gaussian mixtures with a varying number of Gaussians and dimensions.** We plot the test mean squared error (MSE) loss obtained by approximating Gaussian mixtures $f(x; \mu, \Sigma)$ where $\mu \in \mathbb{R}^{N \times d}$ and $\Sigma \in \mathbb{R}^{N \times d \times d}$ using multiplicative and ReLU networks for different values of $d$ and $N$. We report their errors averaged over 20 initialization seeds along with their corresponding 95% confidence intervals.

```
def train_dataloader(self):
    return torch.utils.data.DataLoader(self.train_dataset, batch_size=64, shuffle=True
        )

def val_dataloader(self):
    return torch.utils.data.DataLoader(self.val_dataset, batch_size=32, shuffle=False)

def configure_optimizers(self):
    optimizer = optim.Adam(self.parameters(), lr=1e-3)
    return optimizer
```

## C  Additional Experiments

In this section, we add plots regarding approximation in higher dimensions $d$, shown in Figure 5. We conducted the exact same experiment as in the bottom row of Figure 3 with $N = 500$ and $d = 5, 20, 30$. As can be seen, both architectures are able to achieve near 0 error, yet we still see faster convergence using multiplicative networks over multilayer perceptrons.

## D  Proofs

**Definition 4.** *We say a subset $U$ of a Banach space $V_1$ is embedded in a Banach space $V_2$ if $\|u\|_{V_2} \leq C\|u\|_{V_1}$ for some fixed constant $C$ and for any $u \in U$.*

**Lemma 2.** *Let $r > 0$ and $d \geq 2$. Then, $\mathcal{B}_{2r,2}$ is embedded as a proper subset of $\mathcal{W}^{r,2}$.*

*Proof.* Since $f \in \mathcal{B}_{2r,2}$, we have $f \in L_2(\mathbb{R}^d)$, and therefore, $\|f\|_2 < \infty$. Additionally, by Robert A. Adams (2003)(Theorem 5.2), there exists a constant $C_1 > 0$ such that

$$\|f\|_{\mathcal{W}^{r,2}} \leq C_1(\|f\|_2 + |f|_{r,2}),$$

where $C_1$ is dependant on $r$ and the dimension $d$. Therefore, it remains to prove that $|f|_{r,2}$ is bounded. For this, we use some properties of multivariate function spaces.

We begin with the following claim that we will use in advance. Let $\omega \in \mathbb{R}^d, d \geq 2$, we have

$$|\omega|^{2r} = \|\omega\|_{l_2}^{2r} = \left(\sum_{m=1}^{d} \omega_m^2\right)^r = \sum_{\|\alpha\|_1 = r} \binom{r}{\alpha_1, \ldots, \alpha_d} \cdot (\omega^\alpha)^2, \tag{4}$$

where $\omega^\alpha = \prod_{i=1}^{d} \omega_i^{\alpha_i}$ for $\alpha \in \mathbb{R}^d$. By Jensen's inequality,

$$|f|_{r,2}^2 = \left(\sum_{\alpha:|\alpha|=r} \|D^\alpha f\|_2\right)^2 \leq d^r \sum_{\alpha:|\alpha|=r} \|D^\alpha f\|_2^2$$

By Parseval's Identity in $\mathbb{R}^d$ (Albrecht et al., 1996):

$$\|D^\alpha f\|_2^2 = \frac{1}{(2\pi)^d} \int_{\mathbb{R}^d} |\mathcal{F}(D^\alpha f)(\omega)|^2 \, \mathrm{d}\omega = \frac{1}{(2\pi)^d} \int_{\mathbb{R}^d} |(\mathcal{F}f)(\omega)|^2 |\omega^\alpha|^2 \, \mathrm{d}\omega.$$

Hence, by (4)

$$\sum_{\alpha:|\alpha|=r} \|D^\alpha f\|_2^2 = \sum_{\alpha:|\alpha|=r} \frac{1}{(2\pi)^d} \int_{\mathbb{R}^d} |(\mathcal{F}f)(\omega)|^2 |\omega^\alpha|^2 \, \mathrm{d}\omega$$

$$\leq \frac{1}{(2\pi)^d} \int_{\mathbb{R}^d} |(\mathcal{F}f)(\omega)|^2 \left(\sum_{\alpha:|\alpha|=r} \binom{r}{\alpha_1, \ldots, \alpha_d} \cdot |\omega^\alpha|^2\right) \mathrm{d}\omega$$

$$= \frac{1}{(2\pi)^d} \int_{\mathbb{R}^d} |\mathcal{F}f(\omega)|^2 |\omega|^{2r} \, \mathrm{d}\omega.$$

where the inequality follows from the fact that $\binom{r}{\alpha_1, \ldots, \alpha_d} \geq 1$. Finally, we conclude that

$$\|f\|_{\mathcal{W}^{r,2}} \leq C_1(\|f\|_2 + |f|_{r,2}) \leq C_1\left(\|f\|_2 + \frac{d^r}{(2\pi)^{d/2}} \left(\int_{\mathbb{R}^d} |(\mathcal{F}f)(\omega)|^2 |\omega|^{2r} \, \mathrm{d}\omega\right)^{1/2}\right)$$

$$\leq C_2 \|f\|_{\mathcal{B}_{2r,2}},$$

for some $C_1, C_2 > 0$, where the second inequality is due to the definition of $\|\cdot\|_{\mathcal{B}_{2r,2}}$ as given in equation 1.

To see that $\mathcal{B}_{2r,2}$ is a proper subset of $\mathcal{W}^{r,2}$, we present a prototype function $f \in \mathcal{W}^{r,2}(\mathbb{R}^d)$, such that $f \notin \mathcal{B}_{2r,r}$. Let $\epsilon > 0$ and $d, r$ such that $2r + \epsilon < d$. We define the function $f \in \mathcal{W}^{r,2}(\mathbb{R}^d)$ through its Fourier transform:

$$\mathcal{F}f(\omega) = \begin{cases} 1 & |\omega| \leq 1 \\ |\omega|^{-(r+(d+\epsilon)/2)} & |\omega| > 1. \end{cases}$$

Indeed, $f \in \mathcal{W}^{r,2}(\mathbb{R}^d)$ since:

$$\int_{\mathbb{R}^d} |\omega|^{2r} |\mathcal{F}f(\omega)|^2 \, \mathrm{d}\omega = \int_{|\omega|\leq 1} |\omega|^{2r} |\mathcal{F}f(\omega)|^2 d\omega + \int_{|\omega|>1} |\omega|^{2r} |\mathcal{F}f(\omega)|^2 \, \mathrm{d}\omega$$

$$\leq 1 + \int_{|\omega|>1} |\omega|^{-(d+\epsilon)} \, \mathrm{d}\omega < \infty.$$

We have shown above that $|f|_{r,2}^2 \leq d^r \frac{1}{(2\pi)^d} \int_{\mathbb{R}^d} |\mathcal{F}f(\omega)|^2 |\omega|^{2r} d\omega$, and so $f \in \mathcal{W}^{r,2}(\mathbb{R}^d)$. Let us see that $\mathcal{F}f(\omega) \notin L_1$. Since $2r + \epsilon < d \Rightarrow r + (d+\epsilon)/2 < d$, the following integral diverges

$$\int_{\mathbb{R}^d} |\mathcal{F}f(\omega)| d\omega = \int_{|\omega|<1} |\mathcal{F}f(\omega)| d\omega + \int_{|\omega|>1} |\mathcal{F}f(\omega)| d\omega$$

$$= \int_{|\omega|<1} |\mathcal{F}f(\omega)| d\omega + \int_{|\omega|>1} |\omega|^{-(r+(d+\epsilon)/2)} d\omega$$

$$\leq \int_{|\omega|<1} |\mathcal{F}f(\omega)| d\omega + \int_{|\omega|>1} |\omega|^{-d} d\omega = \infty.$$

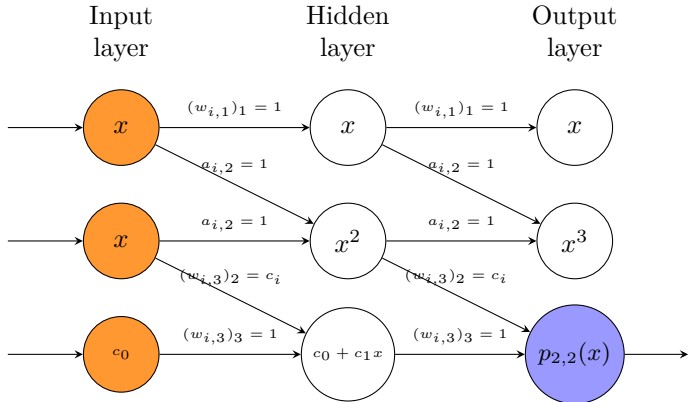

Figure 6: Illustration of the multiplicative network in Proof of Lemma 1, where the polynomial degree $n = 2$.

This implies that $f$ does not satisfy the condition $\frac{1}{(2\pi)^d} \int_{\mathbb{R}^d} |\mathcal{F}f(\omega)| d\omega \leq 1$ which is a necessary requirement for membership in $\mathcal{B}_{2r,2}$. $\qquad\square$

**Lemma 1.** *For any polynomial $p_n : \mathbb{R} \to \mathbb{R}$ of degree $n$ of the form $p_n(x) = \sum_{k=0}^{n} c_k x^k$, there exists a multiplicative neural network $f_n^{\mathrm{POL}} : \mathbb{R}^3 \to \mathbb{R}$, of depth $L_n = \mathcal{O}(n)$ with $G_n = \mathcal{O}(n)$ neurons that satisfies $f_n^{\mathrm{POL}}(x, x, c_0) = p_n(x)$ for $c_0 \in \mathbb{R}$, and $x \in \mathbb{R}$.*

*Proof.* Let $p_n(x) = \sum_{k=0}^{n} c_k x^k$ be a polynomial of degree $n$ and $p_{n,i} = \sum_{k=0}^{i} c_k x^k$ its partial sum up to term $i$. We construct a network $f_n^{\mathrm{POL}}$ whose $i$th layer satisfies:

$$(f_n^{\mathrm{POL}})_i(x, x, c_0) = (x, x^{i+1}, p_{n,i}(x)).$$

We notice that in this construction, the input of the network is a three-dimensional vector composed of the input $x$ and the first coefficient $c_0$. We construct the network by specifying its weights as defined in Definition 2. At layer $i$, the model weights for the three neurons are defined as:

$$w_{i,1} = (1, 0, 0), \quad a_{i,2} = 1 \text{ for } (j_1, j_2) = (1, 2) , \quad w_{i,3} = (0, c_i, 1)$$

The rest of the weights are zeros. In Figure 6, we plot the proposed architecture.

We argue that at any layer $i \geq 0$, neuron 1 contains $x$, neuron 2 contains $x^{i+1}$ and neuron 3 contains $p_{n,i}(x)$. Let $y_{i+1,1}, y_{i+1,2}, y_{i+1,3}$ be the three neurons in layer $i + 1$. Let us address each claim individually. Since the only non-zero weight affecting the first neuron is in $w_{i+1,1}$, $y_{i+1,1} = y_{i,1} = x$ by the assumption. The only non-zero weight affecting the second neuron appears in $a_{i=1,2}$, the weight affecting the multiplicative interaction, and therefore $y_{i+1,2} = y_{i,1} y_{i,2} = x^{i+1}$. Lastly, $y_{i+1,3}$ depends only on $w_{i+1,3}$, and so $y_{i+1,3} = c_{i+1} y_{i,2} + y_{i,3} = p_{n,i}(x) + c_{i+1} x^{i+1} := p_{n,i+1}(x)$. We conclude using the fact that $p_{n,n}(x) = p_n(x)$. The third neuron in the final layer then holds the value $p_n(x)$. $\qquad\square$

**Theorem 2.** *Let $M \geq 1$, $s > 2$, $C_K > 0$ and $\epsilon \in (0, 1)$ be scalars. Then, for any real-valued analytic function $K : [-M, M] \to \mathbb{R}$ that is analytically continuable to the ellipse $E_s^M$ where $|K(x)| \leq C_K$, there exists a deep multiplicative network $f^{\mathrm{MA}} : [-M, M]^3 \to \mathbb{R}$ (MA stands for 'Multiplicative Analytic') of depth $L_\epsilon = \mathcal{O}(\frac{1}{\log_2 s} \log_2 \frac{C_K}{\epsilon})$ with $G_\epsilon = \mathcal{O}(\frac{1}{\log_2 s} \log_2 \frac{C_K}{\epsilon})$ neurons that satisfies $\|f^{\mathrm{MA}}(x, x, x) - K(x)\|_{L^\infty([-M,M])} \leq \epsilon$.*

*Proof.* Let $M \geq 1$, $s > 2$, $C_K > 0$, $\epsilon \in (0, 1)$ and let $K$ be a real-valued analytic function with the required assumptions with respect to $s, C_K$. We wish to construct a network that approximates $K$. As a first step, we approximate $K$ with a polynomial $h_n$ of degree $n \geq 2$ (to be defined later). Then, we realize $h_n$ with a deep multiplicative network as proposed in Lemma 1. Since $K$ satisfies the conditions of Theorem 1, for any integer $n \geq 2$, there exists a polynomial $h_n$ of degree $n$, such that

$$\|h_n - K\|_{L^\infty([-M,M])} \leq \frac{C_K s^{-n}}{s-1}.$$

For the given $\epsilon > 0$, with $s > 2$, we choose and estimate $n$ as follows

$$n := \frac{1}{\log_2 s} \log_2 \frac{C_K}{\epsilon(s-1)} \leq \frac{1}{\log_2 s} \log_2 \frac{C_K}{\epsilon}.$$

This provides the following relation:

$$
\begin{aligned}
\frac{\epsilon(s-1)}{C_K} &= 2^{\log_2(\frac{\epsilon(s-1)}{C_K})} \\
&= 2^{-\frac{1}{\log_2 s}(\log_2(\frac{C_K}{\epsilon(s-1)}))\log_2 s} \\
&= s^{-\frac{1}{\log_2 s}(\log_2(\frac{C_K}{\epsilon(s-1)}))} = s^{-n}.
\end{aligned}
\tag{5}
$$

In particular,

$$\|h_n - K\|_{L^\infty([-M,M])} \leq \frac{C_K s^{-n}}{s-1} \leq \epsilon.$$

Given that $h_n$ is a polynomial of degree $n$, by Lemma 1, there exists $f^{\mathrm{POL}}$ and $c_0 = h_{n,0}$ such that $h_n(x) = f^{\mathrm{POL}}(x, x, c_0)$, and achieves the required bound. $\qquad\square$

Let us now recall Maurey's Theorem (Pisier, 1980-1981; Vitali D. Milman, 1986) which will assist us in the proof of Theorem 3.

**Lemma 3** (Maurey's theorem). *Let $\mathcal{V}$ be a Hilbert space with norm $\|\cdot\|_\mathcal{V}$. Suppose there exists $Q \subset \mathcal{V}$ such that for every $q \in Q$, $\|q\|_\mathcal{V} \leq b$ for some $b > 0$. Then, for every $f$ in the convex hull of $Q$ and every integer $n \geq 1$, there exists a $f_n$ in the convex hull of $n$ points in $Q$ and a constant $c > b^2 - \|f\|_\mathcal{V}^2$ such that $\|f_n - f\|_\mathcal{V}^2 \leq \frac{c}{n}$.*

**Theorem 3.** *Let $\epsilon \in (0, 1)$, $M > 1$, $d \geq 2$, and $K : \mathbb{R} \to \mathbb{C}$ be an analytic kernel that holds the assumptions of Theorem 2 with respect to $s > 2$, $C_K > 0$, and bounded by a constant $D_K \in (0, 1]$ on $[-dM, dM]$. Let $f$ be a real-valued function in $\mathcal{H}_{K,M}(B)$. Further, let $F : [-M, M]^d \to \mathbb{C}$ be a square-integrable function such that $f(x) = \int_{[-M,M]^d} F(\omega) K(\omega \cdot x) \, d\omega$. We define $C_F = \int_{\mathbb{R}^d} |F(\omega)| \, d\omega = \int_{[-M,M]^d} |F(\omega)| \, d\omega$. Then, there exists a deep multiplicative network $f^{\mathrm{MBL}} : B \to \mathbb{R}$ (MBL stands for 'Multiplicative bandlimited') of depth $L_\epsilon = \mathcal{O}\left(\frac{1}{\log_2 s} \log_2 \frac{C_F C_K}{\epsilon}\right)$ with $G_\epsilon = \mathcal{O}\left(\frac{C_F^2}{\epsilon^2 \log_2 s} \log_2 \frac{C_F C_K}{\epsilon}\right)$ neurons that satisfies $\|f^{\mathrm{MBL}} - f\|_{L^2(B)} \leq \epsilon$.*

*Proof.* Let $f \in \mathcal{H}_{K,M}(B)$. Further, let $F(\omega) = |F(\omega)| \cdot \exp(i\theta(\omega))$, the polar representation of $F(\omega)$. We may write $f$ through its Fourier representation:

$$
\begin{aligned}
f(x) &= \mathrm{Re}\left(\int_{[-M,M]^d} F(\omega) K(\omega \cdot x) \, d\omega\right) \\
&= \mathrm{Re}\left(\int_{[-M,M]^d} C_F \exp(i\theta(\omega)) K(\omega \cdot x) \frac{|F(\omega)|}{C_F} \, d\omega\right) \\
&= \int_{[-M,M]^d} C_F [\cos(\theta(\omega)) K_R(\omega \cdot x) - \sin(\theta(\omega)) K_I(\omega \cdot x)] \frac{|F(\omega)|}{C_F} \, d\omega,
\end{aligned}
\tag{6}
$$

where $K_R, K_I$ are the real and imaginary parts of $K$ respectively. Since $\int_{[-M,M]^d} \frac{|F(\omega)|}{C_F} \, d\omega = 1$, equation 6 represents $f$ as an infinite convex combination of functions in

$$Q_{K,M} = \left\{ \gamma[\cos(\beta) K_R(\omega \cdot x) - \sin(\beta) K_I(\omega \cdot x)] \;\middle|\; |\gamma| \leq C_F, \beta \in \mathbb{R}, \omega \in [-M, M]^d \right\}.$$

This means that $f$ is in the closure of the convex hull of $Q_{K,M}$. Due to the fact that $x \in [0, 1]^d$ and $\omega \in [-M, M]^d$, $\omega \cdot x \in [-dM, dM]$. By the definition of $D_K$, functions in $Q_{K,M}$ are bounded in the $L^2(B)$-norm by $2C_F D_K \leq 2C_F$. Using Lemma 3, for any $0 < \epsilon_0 < 1$, to be defined at a later time, there exist real

coefficients $\{b_j\}$ and $\{\beta_j\}$, and vectors $\omega_j \in [-M, M]^d$, $1 \le j \le \lceil 1/\epsilon_0^2 \rceil$, such that:

$$f_{\epsilon_0}(x) = \sum_{j=1}^{\lceil 1/\epsilon_0^2 \rceil} b_j[\cos(\beta_j)K_R(\omega_j \cdot x) - \sin(\beta_j)K_I(\omega_j \cdot x)], \quad \sum_{j=1}^{\lceil 1/\epsilon_0^2 \rceil} |b_j| \le C_F,$$

and

$$\|f_{\epsilon_0} - f\|_{L^2(B)} \le 2C_F\epsilon_0.$$

We are now ready to approximate $f_{\epsilon_0}(x)$ using a deep multiplicative neural network $f^{\mathrm{MBL}}$ (MBL stands for 'Multiplicative bandlimited') on $B$. We notice that $K_R$ and $K_I$ are analytic kernels that satisfy the assumptions of Theorem 2, for $C_K, s, M, \epsilon_0$. They can therefore be approximated to $\epsilon_0$ error using networks $f^{\mathrm{RMA}}$, $f^{\mathrm{IMA}}$ of depth and number of neurons

$$G_{\epsilon_0} \sim L_{\epsilon_0} = \mathcal{O}(\frac{1}{\log_2 s} \log_2 \frac{C_K}{\epsilon_0}),$$

where RMA stands for 'Real Multiplicative Analytic' and IMA stands for 'Imaginary Multiplicative Analytic'. Let us define the multiplicative network $f^{\mathrm{MBL}}(x)$ by

$$f^{\mathrm{MBL}}(x) = \sum_{j=1}^{\lceil 1/\epsilon_0^2 \rceil} b_j\big[\cos(\beta_j)f^{\mathrm{RMA}}(\omega_j \cdot x) - \sin(\beta_j)f^{\mathrm{IMA}}(\omega_j \cdot x)\big].$$

This network has depth $L_{\epsilon_0} = \mathcal{O}(\frac{1}{\log_2 s} \log_2 \frac{C_K}{\epsilon_0})$ and $G_{\epsilon_0} = \mathcal{O}(\frac{1}{\epsilon_0^2 \log_2 s} \log_2 \frac{C_K}{\epsilon_0})$ neurons.

$$\begin{aligned}
\|f^{\mathrm{MBL}}(x) - f_{\epsilon_0}(x)\|_{L^\infty(B)} &\le \sum_{j=1}^{\lceil 1/\epsilon_0^2 \rceil} |b_j| \|f^{\mathrm{RMA}}(\omega_j \cdot x) - K_R(\omega_j \cdot x)\|_{L^\infty(B)} \\
&+ \sum_{j=1}^{\lceil 1/\epsilon_0^2 \rceil} |b_j| \|f^{\mathrm{IMA}}(\omega_j \cdot x) - K_I(\omega_j \cdot x)\|_{L^\infty(B)} \\
&\le 2C_F\epsilon_0,
\end{aligned}$$

that implies

$$\|f^{\mathrm{MBL}}(x) - f(x)\|_{L^2(B)} \le \|f^{\mathrm{MBL}}(x) - f_{\epsilon_0}(x)\|_{L^2(B)} + \|f_{\epsilon_0}(x) - f(x)\|_{L^2(B)} \le 4C_F\epsilon_0,$$

where the last inequality is due to the fact that

$$\|g\|_{L^2(B)} = \left(\int_B |g|^2\right)^{1/2} \le \|g\|_{L^\infty(B)}.$$

Taking $\epsilon_0 = \epsilon/(4C_F)$ achieves the sought result. $\qquad\square$

We further investigate how the constant $C_K$ from Theorem 3 varies as a function of $M$ and $s$. Let $K(u) = \exp(iu)$ be an example kernel, $u \in [-dM, dM]$. We notice that $a_s^{dM} = dM(s + s^{-1})/2$ for $s > 2$, is the larger axis, and therefore the maximal norm of $K$ on $E_s^{dM}$ is given by

$$C_K(s, dM) = \exp(dM\tfrac{s+s^{-1}}{2}).$$

The resulting network $f^{\mathrm{MBL}}$ then has depth

$$L_\epsilon = \mathcal{O}\left(\frac{1}{\log_2 s}\left(dM\frac{s + s^{-1}}{2} + \log_2 \frac{C_F}{\epsilon}\right)\right)$$

and

$$G_\epsilon = \mathcal{O}\left(\frac{C_F^2}{\epsilon^2 \log_2 s}\left(dM\frac{s+s^{-1}}{2} + \log_2 \frac{C_F}{\epsilon}\right)\right)$$

neurons. In this scenario, we see that both a large band $M$ and a large dimension $d$ will linearly affect the first term.

**Theorem 4.** *Let $d \geq 2, r \in \mathbb{N}$, $f \in \mathcal{B}_{2r,2}$ and $\epsilon > 0$. There exists a deep ReLU network $f^{\mathrm{RS}}$ (standing for "ReLU Sobolev") with a depth of $L_\epsilon = \mathcal{O}(d^2\epsilon^{-2/r})$ and $G_\epsilon = \mathcal{O}(d^2\epsilon^{-(2+2/r)})$ neurons, such that $\left\|f^{\mathrm{RS}} - f\right\|_{L_2(B)} \leq \epsilon$.*

*Proof.* Let $f \in \mathcal{B}_{2r,2}$. Similar to the proof of Theorem 5, we define the bandlimiting of $f : \mathbb{R}^d \to \mathbb{R}$ for given $M > 1$ by

$$f_M = \mathcal{F}^{-1}(\mathcal{F}f\mathbb{1}_{[-M,M]^d}).$$

Let us define $F : [-M, M]^d \to \mathbb{C}$:

$$F(\omega) = \frac{1}{(2\pi)^d}(\mathcal{F}f)(\omega),$$

and $K(u) = \exp(iu)$. We then have the following identity:

$$f_M(x) = \int_{[-M,M]^d} F(\omega)K(\omega \cdot x)\,\mathrm{d}\omega = \int_{[-M,M]^d} \tfrac{1}{(2\pi)^d}\mathcal{F}f(\omega)\exp(i\omega \cdot x)\,\mathrm{d}\omega.$$

It is easy to see that $f_M \in \mathcal{H}_{K,M}(B)$. We choose $M := (2/\epsilon)^{1/r}$. Using the same derivations as in the proof of Theorem 5, we seek to approximate $f_M$ with a network $f^{\mathrm{RS}}$ ("ReLU Sobolev") such that

$$\|f_M - f^{\mathrm{RS}}\|_{L_2(B)} \leq \epsilon/2.$$

Once this is achieved, we have:

$$\begin{aligned}
\|f - f^{\mathrm{RS}}\|_{L_2(B)} &\leq \|f - f_M\|_{L_2(B)} + \|f_M - f^{\mathrm{RS}}\|_{L_2(B)} \\
&\leq M^{-r} + \epsilon/2 \leq \epsilon.
\end{aligned} \tag{7}$$

We consider that $C_F = \int_{[-M,M]^d}|F(\omega)|\,\mathrm{d}\omega = \frac{1}{(2\pi)^d}\int_{[-M,M]^d}|\mathcal{F}f(\omega)|\,\mathrm{d}\omega \leq 1$, where the inequality is due to the definition of $\mathcal{B}_{2r,2}$. The kernel $K(u) = \exp(iu)$ takes as input $u = \omega \cdot x$, for $x \in B$ and $\omega \in [-M, M]^d$. Therefore, $u \in [-dM, dM]$, and $K : [-dM, dM] \to \mathbb{R}$. Since $K$ is an entire function, it is continuable to $E_s^{dM}$ for any $s > 2$, let us choose $s = 4$. We notice that $a_4^{dM} = dM(4 + 4^{-1})/2$ is the larger axis, and therefore the maximal norm of $K$ on $E_s^{dM}$ is given by:

$$C_K = \exp(dM\tfrac{4+4^{-1}}{2}).$$

Further, for any $u \in \mathbb{R}$ we have $|K(u)| \leq 1 = D_K$. Using Theorem 3.2 from (Montanelli & Du, 2021) we can construct a deep ReLU network $f^{\mathrm{RS}}$ such that

$$\|f_M - f^{\mathrm{RS}}\|_{L_2(B)} \leq \epsilon/2$$

whose depth is

$$\begin{aligned}
L_\epsilon &\leq C_1\frac{1}{\log_2^2 4}\log_2^2\frac{2C_F C_K}{\epsilon} \\
&\leq C_2\frac{1}{\log_2^2 4}\left(\log_2\frac{\exp(dM\frac{4+4^{-1}}{2})}{\epsilon}\right)^2 \\
&\leq 12C_2\left(d\left(\frac{2}{\epsilon}\right)^{\frac{1}{r}} - \log_2\frac{1}{\epsilon}\right)^2 \\
&\leq C_3 d^2\epsilon^{-\frac{2}{r}},
\end{aligned} \tag{8}$$

for some constants $C_1, C_2, C_3 > 0$. In addition, the number of neurons can be bounded by

$$
\begin{aligned}
G_\epsilon \;&\leq\; C_1' C_F^2 \frac{1}{\epsilon^2 \log_2^2 4} \log_2^2 \frac{2 C_K C_F}{\epsilon} \\
&\leq\; C_2' \frac{1}{\epsilon^2 \log_2^2 4} \left( \log_2 \frac{\exp\!\left(dM^{\frac{4+4^{-1}}{2}}\right)}{\epsilon} \right)^2 \\
&\leq\; \frac{12 C_2'}{\epsilon^2} \left( d\left(\frac{2}{\epsilon}\right)^{\frac{1}{r}} - \log_2 \frac{1}{\epsilon} \right)^2 \\
&\leq\; C_3' d^2 \epsilon^{-(2+\frac{2}{r})},
\end{aligned}
\tag{9}
$$

for some constants $C_1', C_2', C_3' > 0$. $\qquad\square$

**Corollary 1.** *Let $d \geq 2, r \in \mathbb{N}$, $f \in \mathcal{W}^{r,2}$ such that $\|\mathcal{F}f\|_{L_1(\mathbb{R}^d)} < \infty$, and $\epsilon > 0$. There exists a deep ReLU network $f^{\mathrm{MS}}$ (standing for "Multiplicative Sobolev") with a depth of $L_\epsilon = \mathcal{O}(\epsilon^{-1/r})$ and $G_\epsilon = \mathcal{O}(\epsilon^{-(2+(1/r))})$ neurons, such that $\left\| f^{\mathrm{MS}} - f \right\|_{L_2(B)} \leq C \max\{|f|_{r,2}^2, \|\mathcal{F}f\|_1\} \epsilon$, where $C > 0$ is a constant dependent on $r, d$.*

*Proof.* Let $f \in \mathcal{W}^{r,2}$ such that $\mathcal{F}f \in L_1(\mathbb{R}^d)$, and let $\epsilon > 0$. We define a function

$$
\tilde{f} := \frac{f}{d^r \max\{|f|_{r,2}^2 \cdot \|\mathcal{F}f\|_1\}}
$$

We will show that $\tilde{f} \in \mathcal{B}_{2r,2}$ and then by applying Theorem 5 on $\tilde{f}$ we will obtain the desired result.

By the definition of the Sobolev space, $f \in L_2(\mathbb{R}^d)$ and therefore $\tilde{f} \in L_2(\mathbb{R}^d)$. We consider $\mathcal{F}\tilde{f}$,

$$
\begin{aligned}
\frac{1}{(2\pi)^d} \int_{\mathbb{R}^d} |\mathcal{F}\tilde{f}(\omega)| d\omega \;&=\; \frac{1}{(2\pi)^d d^r \max\{|f|_{r,2}^2, \|\mathcal{F}f\|_1\}} \int_{\mathbb{R}^d} |\mathcal{F}f(\omega)| d\omega \\
&=\; \frac{\|\mathcal{F}f\|_1}{(2\pi)^d d^r \max\{|f|_{r,2}^2, \|\mathcal{F}f\|_1\}} \leq 1.
\end{aligned}
$$

We would like to show that $\frac{1}{(2\pi)^d} \int_{\mathbb{R}^d} |\omega|^{2r} |\mathcal{F}\tilde{f}(\omega)|^2 d\omega \leq 1$. Let $\omega \in \mathbb{R}^d, d \geq 2$, we have

$$
|\omega|^{2r} \;=\; \|\omega\|_{l_2}^{2r} \;=\; \left( \sum_{m=1}^{d} \omega_m^2 \right)^r \;=\; \sum_{\|\alpha\|_1 = r} \binom{r}{\alpha_1, \ldots, \alpha_d} \cdot (\omega^\alpha)^2,
\tag{10}
$$

where $\omega^\alpha = \prod_{i=1}^d \omega_i^{\alpha_i}$ for $\alpha \in \mathbb{R}^d$. In addition,

$$
|f|_{r,2}^2 \;=\; \left( \sum_{\alpha:|\alpha|=r} \|D^\alpha f\|_2 \right)^2 \;\geq\; \sum_{\alpha:|\alpha|=r} \|D^\alpha f\|_2^2
$$

By Parseval's Identity in $\mathbb{R}^d$ (Albrecht et al., 1996):

$$
\|D^\alpha f\|_2^2 \;=\; \frac{1}{(2\pi)^d} \int_{\mathbb{R}^d} |\mathcal{F}(D^\alpha f)(\omega)|^2 \, d\omega \;=\; \frac{1}{(2\pi)^d} \int_{\mathbb{R}^d} |(\mathcal{F}f)(\omega)|^2 |\omega^\alpha|^2 \, d\omega.
$$

Hence, by (10)

$$
\begin{aligned}
\sum_{\alpha:|\alpha|=r} \|D^\alpha f\|_2^2 \;&=\; \sum_{\alpha:|\alpha|=r} \frac{1}{(2\pi)^d} \int_{\mathbb{R}^d} |(\mathcal{F}f)(\omega)|^2 |\omega^\alpha|^2 \, d\omega \\
&\geq\; \frac{1}{(2\pi)^d \sum_{\alpha:|\alpha|=r} \binom{r}{\alpha_1,\ldots,\alpha_d}} \int_{\mathbb{R}^d} |(\mathcal{F}f)(\omega)|^2 \left( \sum_{\alpha:|\alpha|=r} \binom{r}{\alpha_1,\ldots,\alpha_d} \cdot |\omega^\alpha|^2 \right) d\omega \\
&=\; \frac{1}{(2\pi)^d d^r} \int_{\mathbb{R}^d} |\mathcal{F}f(\omega)|^2 |\omega|^{2r} \, d\omega,
\end{aligned}
$$

We then arrive at the following inequality:

$$\frac{1}{(2\pi)^d d^r} \int_{\mathbb{R}^d} |\omega|^{2r} |\mathcal{F}f(\omega)|^2 \ \mathrm{d}\omega \le |f|_{r,2}^2.$$

Finally,

$$\begin{aligned}
\frac{1}{(2\pi)^d} \int_{\mathbb{R}^d} |\omega|^{2r} |\mathcal{F}\tilde{f}(\omega)|^2 d\omega &= \frac{1}{(2\pi)^d d^r \max\{|f|_{r,2}^2, \|\mathcal{F}f\|_1\}} \int_{\mathbb{R}^d} |\omega|^{2r} |\mathcal{F}f(\omega)|^2 d\omega \\
&\le \frac{|f|_{r,2}^2}{\max\{|f|_{r,2}^2, \|\mathcal{F}f\|_1\}} \le 1.
\end{aligned}$$

$\square$

