# OpenReview forum: "Exploring the Approximation Capabilities of Multiplicative Neural Networks for Smooth Functions"
_TMLR — Accepted by TMLR_

### Review · Reviewer_TGTp · 2023-02-01

**Summary Of Contributions:**

The authors theoretically analyse the approximation power of multiplicative neural layers compared to standard neural layers, building on previous results that are cited in the paper.

The provided theorems and discussion in this paper extends previous findings to additional types of neural networks, as well as to different types of functions.

**Audience:**

Yes

**Broader Impact Concerns:**

I do not find any ethical issues in this work.

**Claims And Evidence:**

Yes

**Requested Changes:**

Please consider adding an experimental section that shows the difference between the approximation abilities of multiplicative vs standard neural layers.

Please consider adding the following background papers, in the context of using multiplicative layers in learning systems:
 - Stable architectures for deep neural networks
 -  MULTIPLICATIVE INTERACTIONS AND WHERE TO FIND THEM
 - PDE-GCN: Novel Architectures for Graph Neural Networks Motivated by Partial Differential Equations

**Strengths And Weaknesses:**

Strengths:

 - The paper is well organized and easy to follow.

 - The paper is well motivated and gives examples of where a multiplicative layer is used in practice and concerns with types of functions that can be better approximated using a multiplicative layer.

Weaknesses:

- I appreciate the theoretical analysis and the findings in this work. However, I feel that it lacks computational part, that in my opinion should be included in the paper, even in the form of several simple examples. For instance, I think that the paper will benefit from a short experimental section where the authors provide an example function that is hard to approximate using a standard neural layer vs a multiplicative layer.

- The authors may want to add some additional references where multiplicative layers are used in practice (please see below).

Question to the authors:
 - In page 6, you mention "we
do not incorporate non-linearities (ReLU activations) between the intermediate layers".
Can you please elaborate on this point? why does a multiplicative linear has to be linear? is it an assumption of your paper, or is there a fundamental restriction that requires having a linear layer?

Minor comments:

 - Section 2.1: the last two sentences can be merged. they are quite repetitive.

 - Page 5, above equation (1): you may want to directly refer the reader to the appendix section where the Lemma is presented.

---

> ### Author Response · Authors · 2023-02-11
> **Author response**
>
> We would like to thank the reviewer for taking the time to read our paper, and providing useful feedback for our work! Following the review, we uploaded a revised version of the paper with the following changes:
> 1. We clarified the point in page 6.
> 2. We merged the two sentences at the end of Section 2.1.
> 3. We added an explicit reference to the Appendix Section where the lemma is presented.
> 4. We added a discussion of the three papers mentioned by the reviewer (the blue parts in pages 1-2).
> 5. We added Section 4 that describes several experiments that empirically validate our results (in Figures 1-2). For additional experimental details we added Appendix B, including the code used to produce the experiments.

---

> > ### Comment · Reviewer_TGTp · 2023-02-12
> > **Further discussion**
> >
> > I read the revised paper. I thank the authors for the added discussions and experimental section.
> > Especially it is interesting to the see results with varying number of parameters and layers, although I would recommend to experiment with even wider networks, as typical neural networks employ more channels.
> >
> > It is also nice that the authors show that multiplicative neural networks perform better as the frequency of the signal increases.
> >
> > Overall I am happy with the modifications and I therefore revised my review accordingly.

---

### Review · Reviewer_YdBi · 2023-02-20

**Summary Of Contributions:**

This paper establishes approximation guarantees of neural networks. The focus is on the advantage of multiplicative activation functions compared to ReLU activations. In particular, the obtained approximation rates of multiplicative activation improve those of ReLU activation by a factor of dimension or logarithmic terms. In addition to theoretical results, the paper provides empirical study of demonstrating trained neural networks for regression tasks. The MSE of multiplicative networks outperforms ReLU networks, further supporting the theory.

**Audience:**

Yes

**Claims And Evidence:**

Yes

**Requested Changes:**

In Definition 2, what is indexes $j_1, j_2$?

Self-Attention can be seen as an example of multiplicative networks, but the structure of the presentation makes self-attention looks like another network architecture.

Following Definition 3, why taking $s = 2$ corresponds to a circle? If I am not misunderstanding, with $s = 2$, the two semi-axes are not equal.

Following Theorem 2, when comparing to Montanelli et al., (2021), the improvement is a logarithmic factor in terms of the approximation error. Why this leads to an exponentially faster convergence to the target function?

In all the experiments, MSE is plotted of multiplicative and ReLU networks, respectively. However, there is a gap between the MSE and the approximation accuracy of networks. In fact, MSE consists of both bias and variance of the learner model. A smaller MSE not necessarily implies smaller approximation error. This can be made clear in verbal description. Yet the experiments still stand to demonstrate the advantage of multiplicative networks.

Some recent references are missing in introduction. Approximation with structured input space or target function space: "Efficient approximation of deep relu networks for functions on low dimensional manifolds", "Deep Network Approximation in Terms of Intrinsic Parameters", "Deep learning is adaptive to intrinsic dimensionality of model smoothness in anisotropic Besov space". Please also check references therein.

**Strengths And Weaknesses:**

The paper is relatively well organized, with some technical details a bit hard to follow. But this reviewer thinks that it is the complication of the content itself, such as Sobolev function, Bernstein s-ellipse, etc.

Most part of the results seem correct. To better highlight the contribution, it is suggested that some key technical steps in proofs can be explained.

====================== Weaknesses ========================

The target function classes studied in the paper is purely technical. On the one hand, the function class is relatively small so that whose approximation does not suffer from the curse of dimensionality. On the other hand, these function classes give the impression that they are studied because people know how to approximate them.

Compared to ReLU networks, multiplicative networks are automatically capable of exactly implementing polynomials without any error. From a theoretical standpoint, the approximation theory of using multiplicative networks seem to be thin and the advantage over ReLU networks is highly tied to the activation function. This limits the contribution of the paper.

A minor point is there is no lower bound established. Are the obtained approximation rates optimal?

---

> ### Author Response · Authors · 2023-02-26
> **Author response**
>
> We would like to thank the reviewer for taking the time to read our paper, and providing useful feedback for our work! Following the review, we uploaded a revised version of the paper (marked as red) with the following changes.
>
> R: The target function classes studied in the paper is purely technical. On the one hand, the function class is relatively small so that whose approximation does not suffer from the curse of dimensionality. On the other hand, these function classes give the impression that they are studied because people know how to approximate them.
> > A: We address this section in the review in two manners.
> 1. We add a reference to an example in the appendix, showing that the integrability condition of Fourier transform is important, and does not hold for all functions in ${W}^{r, 2}$ (see the end of the proof of Lemma 2).
> 2. We added Corollary 1 showing that one can approximate members of the Sobolev space $W^{r,2}$ with favorable complexity guarantees when using multiplicative neural networks, assuming that the Fourier transform of the function is integrable.
>
> R: Most part of the results seem correct. To better highlight the contribution, it is suggested that some key technical steps in proofs can be explained.
> >A: To strengthen their clarity, we have included additional technical details to the proofs (see the red parts throughout the proofs).
>
> R: In Definition 2, what is indexes $j_1, j_2$?
> >A: We added an explicit reference to the indices in Definition 2. Furthermore, we extended the discussion regarding the differences between multiplicative layers and fully-connected layers (the red part after Definition 2). These differences, although minor, provide better approximation guarantees, as reflected in the results.
>
> R: Self-Attention can be seen as an example of multiplicative networks, but the structure of the presentation makes self-attention looks like another network architecture.
> >A: It is indeed true that specific parts of the self-attention layer may be introduced as multiplicative layers. However, we have attempted to present the self-attention as presented in (Vaswani et al. 2017), which incorporates other mechanisms, such as residual connections, linear projections, and ReLU activations (as discussed in Section 2.2).
>
> R: Following Definition 3, why taking s=2 corresponds to a circle? If I am not misunderstanding, with j=2, the two semi-axes are not equal.
> >A: We thank the reviewer for pointing out this mistake, we removed this comment.
>
> R: Following Theorem 2, when comparing to Montanelli et al., (2021), the improvement is a logarithmic factor in terms of the approximation error. Why this leads to an exponentially faster convergence to the target function?
> >A: We thank the reviewer for finding this mistake, we fixed this statement following Theorem 2.
>
> R: In all the experiments, MSE is plotted of multiplicative and ReLU networks, respectively. However, there is a gap between the MSE and the approximation accuracy of networks. In fact, MSE consists of both bias and variance of the learner model. A smaller MSE not necessarily implies smaller approximation error. This can be made clear in verbal description. Yet the experiments still stand to demonstrate the advantage of multiplicative networks.
> >A: We agree with the reviewer’s comment regarding estimating the approximation error. This is indeed in inherent issue that arises when conducting experiments to estimate approximation errors. Following the review, we added a discussion in Section 4 on common practices to overcome this limitation and emphasize our approach for dealing with this issue.
>
> R: Some recent references are missing in introduction. Approximation with structured input space or target function space: "Efficient approximation of deep relu networks for functions on low dimensional manifolds", "Deep Network Approximation in Terms of Intrinsic Parameters", "Deep learning is adaptive to intrinsic dimensionality of model smoothness in anisotropic Besov space". Please also check references therein.
> >A: We added the aforementioned references, along with other relevant papers that prove approximation guarantees of neural networks when the data lies on a low dimensional space, or the function has as particular structure (see the red text in Section 1).

---

> > ### Author Response · Authors · 2023-04-20
> > **Updated visibility permission for author response**
> >
> > We sincerely apologize for the error in the previous reply (sent on February 26, 2023) which was sent with incorrect permissions (Chief, Action Editors, Authors) and may not have been visible to the reviewer. We are grateful for the reviewer's valuable time and have incorporated all the requested changes in the manuscript. If there are any further concerns that need to be addressed, we will gladly do so in the revised version.

---

### Review · Reviewer_hvgK · 2023-03-03

**Summary Of Contributions:**

The paper considers the multiplicative neural networks where instead of using non-linear activation function at each layer, the neurons are connected with simple multiplication operations. The authors explores how well the multiplicative neural networks can approximate smooth functions on Sobolev space as well as band-limited smooth functions. The authors show that compared to the standard ReLU neural networks, with the same error tolerance, the multiplicative NNs can approximate smooth or bandlimited functions with fewer number of layers or fewer neurons.
To achieve these results, the authors first show that multiplicative NNs can efficiently represent polynomial functions and then use the existing results on approximating functions with polynomials or kernels to achieve the desired results.
This potentially can shed some light on the success of the attention blocks and transformers as these models have some forms of multiplicative layers in their architecture.

**Audience:**

Yes

**Claims And Evidence:**

Yes

**Requested Changes:**

The authors should consider adding simulations to support their theoretical results (Thm. 2-4) which I believe can considerably improve the quality of the paper.

Moreover, there are a few typos or ambiguities in the manuscript:
1. Problem setup, first equation: This formulation implies that a single best approximating function from H is used to approximate all functions in U. It seems that the authors meant to consider a separate approximator for each function in U.
2. Definition 2: worth to mention that $j_1$ and $j_2$ can be any arbitrary index depending on $j$ and $i$.
3. The paragraph after definition 3 is incorrect. Note that as $s\rightarrow \infty$, the Bernstein s-ellipse converges to a circle. As $s$ decreases, the ellipse becomes more elongated. Moreover, it is not necessary to limit $M\geq1$. $M>0$ would be enough.
4. proof of Thm. 5, equation 2, the reason for the third inequality is not clear.
5. Experiments section: to avoid bad local minima, did the authors run their simulations multiple times with different seeds? If so, how many times? otherwise, to have more stable results, would be good to reiterate the simulations multiple times and show the mean and variance of the MSE.
6. The line after equation 4, the definition of $\omega^{\alpha}$ should be corrected as $\omega^{\alpha}=\Pi_{i=1}^d \omega_i^{\alpha_i}$.
7. proof of Thm. 3, last equation, duplicate $||g||_{L^{\infty}(B)}$?

**Strengths And Weaknesses:**

Strengths:
The paper is well-written and the results are clearly stated and compared to the existing methods. The analysis, to my knowledge, is novel and the improvement over traditional ReLU networks is noteworthy.

Weaknesses:
The only major weakness is the simulations which are limited to Chebyshev polynomials which can be interpreted as simple validation of Lemma 1. It is expected to see simulations verifying the main theorems of the paper.

---

> ### Author Response · Authors · 2023-03-11
> **Author response**
>
> We would like to thank the reviewer for taking the time to read our paper, and providing useful feedback for our work! Following the review, we uploaded a revised version of the paper with the following changes (marked as brown).
>
> Reviewer: The only major weakness is the simulations which are limited to Chebyshev polynomials which can be interpreted as simple validation of Lemma 1. It is expected to see simulations verifying the main theorems of the paper.
>
> > Following the review, we added new experiments comparing the approximation capabilities of multiplicative networks and standard multilayer perceptrons for approximating multivariate mixture of Gaussians, which are members of the space $\mathcal{B}_{2r,2}$ up to a constant factor. These experiments, which are described in Section 4 (marked as brown) and are provided in Figure 3 demonstrate that multiplicative networks are able to achieve better approximation errors when compared with multilayer perceptrons.
>
> Reviewer: Problem setup, first equation: This formulation implies that a single best approximating function from $\mathcal{H}$ is used to approximate all functions in $\mathcal{U}$. It seems that the authors meant to consider a separate approximator for each function in $\mathcal{U}$.
>
> > Following the review, we fixed this mistake.
>
> Reviewer: Definition 2: worth to mention that $j_1, j_2$  can be any arbitrary index depending on $i, j$.
>
> > We added an explicit reference to the indices in Definition 2 and mentioned their potential dependence on $i,j$.
>
> Reviewer: The paragraph after definition 3 is incorrect. Note that as s\rightarrow \infty, the Bernstein s-ellipse converges to a circle. As s decreases, the ellipse becomes more elongated. Moreover, it is not necessary to limit M≥1. M>0 would be enough.
>
> > Following the review, we fixed these issues.
>
> Reviewer: Proof of Thm. 5, equation 2, the reason for the third inequality is not clear.
>
> > To clarify equation 2 we included additional steps in the analysis and extended the text below the equation to improve the discussion of them.
>
> Reviewer: Experiments section: to avoid bad local minima, did the authors run their simulations multiple times with different seeds? If so, how many times? otherwise, to have more stable results, would be good to reiterate the simulations multiple times and show the mean and variance of the MSE.
>
> > In the previous version of the paper, we estimated the approximation error of a given model in approximating a specific function using multiple initialization seeds, and then selected the best-fitting candidate, as described in the red text in Section 4. This approach helped us to avoid bad local minima, and provide a more reliable estimate of the approximation error. Following the reviewer’s suggestions, in our new experiments (Figure 3), we averaged the results over 20 initialization seeds and plotted the results along with their corresponding 95% confidence intervals.
>
> Reviewer: The line after equation 4, the definition of $\omega^\alpha$ should be corrected
> proof of Thm. 3, last equation, duplicate?
>
> > Following the review, we have made the correction to the definition of $\omega^\alpha$ following Equation 4 as suggested and removed the duplicate expression in the last equation of the proof of Theorem 3.

---

### Author Response · Authors · 2023-05-01
**Camera-ready Submission**

We are pleased to inform that we have uploaded the camera-ready revision. We have incorporated the requested changes, and further fixed minor typos in the manuscript. Please let us know if you have any questions or concerns. We thank the reviewers and the area chair for their continual support and constructive feedback in the reviewing process of the manuscript!

---

### Decision · Action_Editors · 2023-04-18

**Recommendation:** Accept with minor revision

**Comment:**

The paper studies an important problem and makes interesting theoretical contributions.

The paper was reviewed by three expert reviewers. Even though two reviewers are satisfied with the author's response, one reviewer still had concerns even after the discussion period. As such, I recommend "Accept with revision", which would provide additional time for authors to address the concerns raised by this reviewer.

In particular, the reviewer is not satisfied with authors' response; yet, they acknowledge that some of their concerns are addressed in the revision. The reviewer's major concern is that the function classes do not show a clear connection to practice.

I recommend authors to explore some practical settings/examples to demonstrate their results. After this change, the paper will be suitable for publication at TMLR.

Best, AC


**Audience:**

The paper study expressive power of neural networks, which is certainly of interest to TMLR community.

**Claims And Evidence:**

The authors study multiplicative neural networks where the neurons are connected with simple multiplication operations. In contrast,  the standard formulation would require non-linear activation function at each layer.

Authors study the approximation properties of these multiplicative networks. They specifically consider smooth functions on Sobolev spaces and band-limited functions. The result proves that multiplicative NNs can approximate these functions  using less number of layers.

---

> ### Author Response · Authors · 2023-04-20
> **Acknowledgment and Apology for Non-Visible Response to a Reviewer**
>
> We extend our sincerest gratitude to the Reviewers and the AC for their insightful reviews and commitment to the quality of the manuscript. We are delighted to learn that the paper has been accepted with minor revision!
>
> We have discovered an error in the response to the review, where the feedback intended for Reviewer "YdBi" was not visible to all reviewers. We apologize for any inconvenience caused and express our appreciation for the time taken by the reviewers to read the paper and provide constructive feedback. We have incorporated all the necessary changes and suggestions in the current version of the revised manuscript, and we have adjusted the visibility settings to ensure that all responses are visible to the reviewers.
>
> We are pleased to inform you that we will soon upload the camera-ready version with further corrections as requested by the AC. We are committed to ensuring the highest quality of our manuscript and will gladly address any further questions or concerns that may arise. Thank you for your continued support and feedback.